



# Algorithm evaluation for Polarimetric Remote Sensing of Atmospheric Aerosols

Otto Hasekamp[1], Pavel Litvinov[2], Guangliang Fu[1], Cheng Chen[2], and Oleg Dubovik[3]

[1]SRON Netherlands Institute for Space Research, Niels Bohrweg 4, 2333 CA Leiden, the Netherlands
[2]GRASP-SAS, Villeneuve-d'Ascq, France
[3]Laboratory of Atmospherioc Optics, Univ. Lille, CNRS, UMR 8518 - LOA - Laboratoire d'Optique Atmosphérique, Lille, France

**Correspondence:** Otto Hasekamp (O.Hasekamp@sron.nl)

**Abstract.** From a passive satellite remote sensing point-of-view, the richest set of information on aerosol properties can be obtained from instruments that measure both intensity and polarization of back- scattered sun light at multiple wavelengths and multiple viewing angles for one ground pixel. However, it is challenging to exploit this information at a global scale because complex algorithms are needed with many fit parameters (aerosol and land/ocean reflection), based on online radiative transfer

models. So far, two of such algorithms have demonstrated capability at a global scale: the Generalized Retrieval of Atmosphere and Surface Properties (GRASP) algorithm and the Remote Sensing of Trace gas and Aerosol Products (RemoTAP) algorithm. In this paper, we present a detailed comparison of the most recent versions of RemoTAP and GRASP. We evaluate both algorithms for synthetic observations, for real PARASOL observations against AERONET for common pixels, and for global PARASOL retrievals for the year 2008. Both RemoTAP and GRASP show good agreement against AERONET. For the Aerosol

Optical Depth (AOD) over land, both algorithms show a Root Mean Square Error (RMSE) of 0.10 (at 550 nm). For Single Scattering Albedo (SSA), both algorithms show a good performance in terms of RMSE ( 0.04) but RemoTAP has a smaller bias (0.002) compared to GRASP (0.021). For Angstrom Exponent (AE), GRASP has a smaller RMSE (0.367) than RemoTAP (0.387), mainly caused by a small overestimate of AE at low values (large particles). Over ocean both algorithms perform very well. For AOD, RemoTAP has an RMSE of 0.057 and GRASP an even smaller RMSE of 0.047. For AE, the RMSE

of RemoTAP and GRASP are 0.285 and 0.224, respectively. Based on the AERONET comparison, we conclude that both algorithms show very similar overall performance, where both algorithms have stronger and weaker points. For the global data products, we find a Root Mean Square Difference (RMSD) between RemoTAP and GRASP AOD of 0.12 and 0.038 over land and ocean, respectively. The largest differences occur over the biomass burning region in equatorial Africa. The global mean values are virtually unbiased with respect to each other. For AE the RMSD between RemoTAP and GRASP is 0.33

over land and 0.23 over ocean. For SSA, we find good agreement over land (RMSD=0.043) for retrievals with AOD > 0.2. Over ocean the agreement is poor with a bias of 0.053 (where RemoTAP retrieves higher SSA) and an RMSD of 0.074. As expected, the differences increase towards low AOD, both over land and ocean. We also compared the GRASP and RemoTAP AOD and AE products against MODIS. For AOD over land, the agreement of either GRASP or RemoTAP with MODIS is worse than the agreement between the 2 PARASOL algorithms themselves. Over ocean, the agreement is very similar among

the 3 products for AOD. For AE, the agreement between GRASP and RemoTAP is much better than the agreement of both

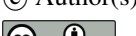



products with MODIS. Overall, the good agreement between RemoTAP and GRASP, and between the individual algorithms with AERONET, demonstrates high fidelity of both data products for PARASOL. The agreement of the latest product versions with each other and with AERONET improved significantly compared to the previous version of the global products of GRASP and RemoTAP. The results demonstrate that dedicated effort on algorithm development for MAP aerosol retrievals still leads to substantial improvement of the resulting aerosol products and this is still an ongoing process.

# 1   Introduction

Terrestrial aerosol is a complex mixture of liquid, solid, or mixed-phase particles emitted by natural and anthropogenic sources. Aerosols affect Earth's climate as they scatter and absorb solar radiation and act as condensation nuclei for cloud droplets and ice crystals. Substantial changes in anthropogenic aerosol emissions in the industrial age occurred. The overall increase in aerosol emissions have led to a cooling of Earth's atmosphere, compensating part of the temperature increase imposed by anthropogenic greenhouse gas emissions (Andreae et al., 2005). Prevailing uncertainty in the (effective) radiative forcing (ERF) from anthropogenic aerosol emissions continue to limit the accuracy of estimates of global climate sensitivity to changes in greenhouse gas concentrations, as clearly stated in the 6th assessment report by the Intergovernmental Panel on Climate Change (IPCC-AR6, Forster and et al. (2021)). This uncertainty severely hampers future predictions of climate change. There have been significant changes in aerosol emissions in the past 2 decades. Over Europe and the USA aerosol emissions have declined since the 1990s. Over East Asia, there has been an increase in aerosol emissions until 2010 and a decline afterwards. Over south Asia, aerosol emissions are continuing to increase (Forster and et al., 2021). Overall, reductions in global aerosol emissions are expected in the next decades, improving air quality, but also likely leading to a further warming of the atmosphere. For Aerosol-Radiation Interactions (ARI) substantial uncertainties exist related to insufficient knowledge on aerosol absorption (Li et al., 2022a). However, Aerosol-Cloud Interactions (ACI) represent the most uncertain contribution to the total aerosol ERF (Bellouin et al., 2020). By acting as Cloud Condensation Nuclei (CCN), aerosols affect the cloud droplet number concentration (Nd) and consequently the cloud albedo (Twomey, 1974; Quaas et al., 2020), causing the Radiative Forcing due to ACI (RFaci). Subsequently, rapid adjustments take place in e.g. Cloud Fraction and Liquid Water Path that result from an initial change in Nd (Gryspeerdt et al., 2020). A key step in quantifying RFaci (and subsequent rapid adjustments) is to quantify the sensitivity of Nd to the number concentration of cloud active aerosol, which depends mostly on the (dry) size distribution (Dusek et al., 2006; Hasekamp et al., 2019a).

To improve our understanding of the effect of aerosols on climate, weather and air quality, measurements of aerosol chemical composition, size distribution, refractive index, optical properties like Aerosol Optical Depth (AOD) and Single Scattering Albedo (SSA), as well as the aerosol height profile are of crucial importance. It has been demonstrated by various studies that Multi-Angle Polarimetric (MAP) measurements are needed to provide information about detailed aerosol properties like size



distribution, refractive index, SSA, in addition to the AOD. The only MAP instrument that has provided a multi-year data set (2005-2013) in the past has been the French POLDER-3 instrument on the PARASOL mission (hereafter simply referred to as PARASOL). Now space agencies realize the large potential of MAP instrumentation (Dubovik et al., 2019), in the 2020s
several of such instruments will be launched, e.g. SPEXone (Hasekamp et al., 2019a) and HARP-2 on PACE by NASA in 2024 (Werdell et al., 2019), 3MI on METOP-SG by ESA in 2025 (Fougnie et al., 2018), a MAP on the CO2-Monitoring mission by ESA in 2026, and a MAP on the AOS mission by NASA in 2028. To cope with the increased information content on aerosols of MAP instrumentation and to assess the climatic effect of aerosols, new tools for retrieval need to be (further) developed. So far, this development has lagged behind the instrument development, which is the reason for the under-exploitation of the
existing PARASOL data sets.

There are currently a number of aerosol retrieval algorithms available for the exploitation of MAP measurements. The first algorithms to generate the operational PARASOL products were using Look-Up Tables (LUTs) based on a limited number of standard aerosol models (a combination of size distribution and refractive index) and omit measurements that have a considerable contribution from land/ocean reflection (Deuzé et al., 2001; Herman et al., 1997). Retrievals from PARASOL
measurements using LUT approaches have so far hardly demonstrated added value compared to single-viewing-angle radiometer retrievals from e.g. MODIS. In order to make full use of the information contained in MAP measurements, full inversion approaches are needed that consider a continuous space of aerosol microphysical properties (size distribution, refractive index), instead of using standard aerosol models, and to properly account for land or ocean reflection by retrieving land or ocean parameters simultaneously with aerosol properties. Examples of such algorithms are the Generalized Retrieval of Aerosol and
Surface Properties (GRASP) algorithm (Dubovik et al., 2011, 2014, 2021; Chen et al., 2020), the Remote Sensing of Trace gas and Aerosol Products (RemoTAP) algorithm (Hasekamp et al., 2011; Fu and Hasekamp, 2018; Fu et al., 2020; Lu et al., 2022), the Jet Propulsion Laboratory (JPL) algorithm (Xu et al., 2017; Xu et al., 2018), the microphysical aerosol properties from polarimetry (MAPP) algorithm (Stamnes et al., 2018), and the Multi-Angular Polarimetric Ocean coLor (MAPOL) algorithm (Gao et al., 2021a, b, 2022). Of the full inversion approaches only the RemoTAP and GRASP algorithms have demonstrated
capability at a global scale. These are the two algorithms are evaluated in the present work.

Already important scientific advancement has been made based on retrieval products from the GRASP and RemoTAP algorithms. For example, data assimilation / inverse modeling studies have been performed for GRASP (Chen et al., 2018, 2019) and RemoTAP (Tsikerdekis et al., 2021, 2023), making use explicitly of unique MAP information on size and absorption by aerosols. Also, the data products have been used for model evaluation of aerosol absorption and quantifying the direct radiative
effect of aerosols (Lacagnina et al., 2016, 2017; Chen et al., 2022b), as well as the radiative forcing due to aerosol cloud interactions (Hasekamp et al., 2019b). Both algorithms have already shown good performance against ground based measurements from the Aerosol Robotic Network (AERONET) (Chen et al., 2020; Lacagnina et al., 2017; Schutgens et al., 2021; Zhang et al., 2021), although based on theoretical information content a better agreement is expected.

In this paper, we present a detailed comparison of the most recent versions of RemoTAP and GRASP. We evaluate both
algorithms for synthetic observations, for real PARASOL observations against AERONET for common pixels, and for global PARASOL retrievals for the year 2008. Through this comparison, we identify aspects of both data products with large fidelity





and aspects that need improvement. Section 2 describes the data and methods used in this study, section 3 shows a comparison between the RemoTAP and GRASP forward models and a comparison for retrieval on simulated observations, section 4 shows a comparison of the GRASP and RemoTAP data products against AERONET and each other. Finally, section 5 concludes the paper.

## 2  Methodology

In this section, we summarize the main characteristics of the RemoTAP and GRASP algorithms, describe experiments with synthetic data, and the validation and comparison approach for retrievals on real PARASOL observations.

### 2.1  Retrieval Algorithms

The RemoTAP algorithm has been developed at SRON Netherlands Institute for Space Research. It combines the capability for aerosol retrieval from MAP measurements (Hasekamp et al., 2011; Wu et al., 2015, 2016; Fu and Hasekamp, 2018; Fu et al., 2020) with the capability to retrieve trace gas columns from spectrometer measurements in the Near-Infrared and Shortwave-Infrared (Hasekamp and Butz, 2008; Butz et al., 2009; Butz et al., 2011) into one algorithm (Lu et al., 2022). The latest version of RemoTAP is described by Lu et al. (2022) for retrievals over land, whereas the ocean model for retrievals over ocean is described by Fan et al. (2019). RemoTAP is based on iterative fitting a linearized radiative transfer model to the measurements of intensity and polarization of light reflected by the Earth atmosphere and surface. It has large flexibility in the definition of parameters to be retrieved and allows retrievals over land, ocean, and clouds. The RemoTAP software has strong heritage in application to PARASOL measurements, airborne measurements of the Research Scanning Polarimeter (RSP), SPEX airborne, and airMSPI. RemoTAP is also planned to be used for operational processing of the SPEXone/PACE data. Main characteristics of the algorithm are summarized in Appendix A.

The GRASP algorithm has been developed and maintained by the Laboratoire d'Optique Atmosphérique (LOA)/CNRS of University of Lille and the GRASP-SAS company. GRASP is a new-generation algorithm developed for deriving extensive aerosol and surface properties from diverse space-borne and ground-based instruments as well as their synergy (Dubovik et al., 2014, 2019, 2021). In applications for MAP the algorithm retrieves a set of aerosol and surface parameters simultaneously using the measurements at all wavelengths, all angles and all states of polarization. GRASP retrieval is based on statistical optimization in frame of Multi-Term Least Square (LSM) concept (Dubovik et al., 2021) that allows the use of multiple constraints simultaneously. In addition, GRASP is designed to perform multi-pixel inversion where statistically optimized retrieval is implemented for a large group of satellite pixels (Dubovik et al., 2011). This allows to improve retrieval by using additional a priori constraints on spatial or temporal variability of any retrieved parameter in different pixels. Moreover, the GRASP algorithm is highly flexible and versatile in modelling of aerosol and surface properties and can be adapted to various situations with different information content in the measurements (Dubovik et al., 2021; Litvinov et al., 2011; Lopatin et al., 2013, 2021; Chen et al., 2020, 2022a; Li et al., 2022b). The open-source software and documentation are available at www.grasp-open.com. Main characteristics of the algorithm are summarized in Appendix A.





Both RemoTAP and GRASP algorithms are so-called 'full physics' algorithms that iteratively fit a radiative transfer model

to observations of intensity and polarization of scattered/reflected light. However, there are also important differences between RemoTAP and GRASP:

- **Inversion approach**: GRASP uses a multi-pixel approach where multiple ground-pixels (in time and space) are used at the same time in the inversion procedure (Dubovik et al., 2011, 2014, 2021), imposing smoothness constraints on the temporal and spatial changes in surface- and aerosol properties. RemoTAP uses a single pixel approach where a separate

inversion is applied for each ground pixel.

- **State vector definition**: GRASP uses an aerosol description based on 5 fixed size modes (3 for the fine and 2 for the coarse mode).The GRASP refractive index for each mode is represented as internal mixture of several chemical components Li et al. (2019). Here, the refractive index for fine mode is represented as internal mixture of the black and brown carbon (BC and BrC), quartz (fine mode dust) and soluble components. The refractive index of the coarse mode is

internal mixture of iron oxides, quartz and soluble components. For both modes the soluble component mimics sulfate, ammonium nitrate or sea salt dissolved in water. RemoTAP uses an aerosol description based on 3 modes where effective radius $r_{\mathrm{eff}}$ and effective variance $v_{\mathrm{eff}}$ of each mode are retrieved. As in GRASP, the refractive index is described by a mixture of chemical components but the contribution of each component is retrieved independently.

- **Forward Model** Both GRASP and RemoTAP use the same pre-calculated tables (kernels) of the aerosol single scattering

optical properties, based on Mie, T-Matrix and geometrical optics computations for spherical and spheroidal particles in a wide range of the complex refractive indices, non-sphericity and size parameters (Dubovik et al., 2006). The kernels are presented on a fixed grid of the size parameter ($\frac{2\pi r}{lambda}$) taken at the wavelength of 340 nm. To do calculations for an arbitrary wavelength, RemoTAP uses a fixed grid of the size parameter so that the tabulated values do not have to be interpolated (but the size distribution has to be computed for a different radius grid for different wavelengths). GRASP

on the other hand uses a fixed radius grid of the size distribution which means that the tabulated optical properties have to be interpolated for each different wavelength. The radiative transfer solvers used by RemoTAP (Landgraf et al., 2002; Hasekamp and Landgraf, 2002; Hasekamp et al., 2004; Schepers et al., 2014) and GRASP (Lenoble et al., 2007; Waquet and Herman, 2019; Herreras-Giralda et al., 2022) are also different but this has minor impact.

- **Measurement vector**: RemoTAP includes (sun-normalized) intensity $I$ together with the Degree of Linear Polarization

(DoLP) which is defined as $\frac{\sqrt{Q^2+U^2}}{I}$. The assumed uncertainty is 1% on $I$ and 0.007 (absolute) on DoLP. GRASP includes includes (sun-normalized) intensity $I$ together with the Stokes fractions q = Q/I and u = U/I.The assumed uncertainty is 1 % on $I$ and 0.002 (absolute) on q and u.

- **Data filtering**: GRASP applies cloud screening from PARASOL based on Zeng et al. (2011) prior to the retrievals and applies a posterior filter removing retrievals with a pixel-level minimum relative fit residual > 3% over land and >

10% over ocean. RemoTAP applies a cloud screening based on MODIS, keeping only cloud fractions < 0.20, prior to





the retrieval and applies a posterior filter removing retrievals with $\chi^2 > 5$. The posterior filter based on $\chi^2$ provides an additional cloud filter (Stap et al., 2015).

The latest version of RemoTAP (Lu et al., 2022) used in this paper is substantially different from the previous version used for global PARASOL processing (Lacagnina et al., 2017). Most important difference with the previous version is the 3-mode aerosol description compared to the 2-mode description used previously. Other differences include a more extended Lookup Table for 1st guess retrieval, the description of the refractive index by a contribution from different chemical components, and the use of new model for the ocean body contribution (Fan et al., 2019). The version of GRASP is based on the recently developed Chemical Component (CC) approach (Li et al., 2019; Zhang et al., 2021), while the previous global versions of GRASP (Chen et al., 2020) were based on the GRASP "Optimized", "High Precision", and "Models" approaches, referred to as GRASP-O, GRASP-HP, and GRASP-M. The validation results of Chen et al. (2020) suggest that GRASP-M provide the most accurate AOD while the most accurate detailed aerosol microphysical properties are provided by GRASP-HP. At the same time, the total AOD from GRASP/HP product had an issue of non-negligible biases for low AOD cases, which suggested a general difficulty to retrieve many free parameters when the information content is low (Chen et al., 2020). In that respect, the latest GRASP/Components approach uses a reduced set of parameters, but more parameters than in the GRASP/Models approach. In the present paper we also shortly summarize the validation results of the previous RemoTAP and GRASP products mentioned above in order to compare with the latest versions.

## 2.2 Setup for Synthetic Comparison

For a comparison between the forward models of RemoTAP and GRASP we use a set of aerosol properties of Table 1 and surface properties of Table 2. We have chosen a typical bi-modal size distribution where we vary the AOD of the fine- and coarse mode respectively. For the geometries, we use PARASOL geometries from overpasses over the AERONET stations of Table 3. The next step is to compare synthetic retrievals of RemoTAP and GRASP. For this purpose we created 2 sets of synthetic measurements that are as realistic as possible and not consistent with assumptions made in both retrieval codes:

- **RemoTAP-ECHAM** Based on aerosol microphysical properties from simulations by the ECHAM-HAM aerosol climate model. ECHAM-HAM provides mass-mixing ratio in different vertical layers of the atmosphere of different aerosol species (Sulfate, Organic Carbon, Black Carbon, Dust, Sea Salt) in seven different size modes: Nucleation Soluble (NS), Aitken Soluble (KS), Accumulation Soluble (AS), Coarse Soluble (CS), Aitken Insoluble (KI), Accumulation Insoluble (AI), Coarse Insoluble (CI). Based on the composition we can compute the refractive index for each mode. The 7 modes aerosol description is different from the standard setup of the RemoTAP algorithm (3 modes with different refractive index) and also from the standard GRASP algorithm (5 modes with a mixture of different chemical components for the refractive index). For surface BRDF / BPDF properties and AOD we use PARASOL retrieved values over the AERONET sites of Table 3. The solar and viewing geometries are based on PARASOL geometries for overpasses over these AERONET stations. The corresponding measurements are generated by the RemoTAP forward model.





- **GRASP-AERONET** Based on AERONET retrievals for the 5 stations of Table 3. AERONET retrieves the aerosol
  size distribution in 22 bins, and one size independent, wavelength dependent complex refractive index. The 22-bin size
distribution is inconsistent with the assumptions in both RemoTAP and GRASP. For surface BRDF/BPDF and geometry
  we use the values from PARASOL overpasses as above. The corresponding measurements are generated by the GRASP
  forward model.

We apply both RemoTAP and GRASP to both sets of synthetic measurements.

**Table 1.** Aerosol properties used for forward model comparisons

| Parameter | Mode 1 (log-normal) | Mode 2 log-normal |
|---|---|---|
| $r_{eff}$ ($\mu$m) | 0.15 | 1.5 |
| $v_{eff}$ | 0.2 | 0.6 |
| RRI | 1.53 | 1.53 |
| IRI | 0.005 | 0.005 |
| $f_{sphere}$ | 1.0 | 0.5 |
| AOD | 0.3, 1.0 | 0, 0.1, 0.5, 1.0, 2.0 |
| Altitude distribution | homogeneously 0-2 km | homogeneously 0-2 km |

**Table 2.** Surface parameters used for the forward model comparison

| Wavelength (nm) | 440 | 490 | 563 | 670 | 865 | 1020 |
|---|---|---|---|---|---|---|
| Ross-Li scaling | 0.03 | 0.03 | 0.05 | 0.07 | 0.40 | 0.5 |
| Ross Thick | 0.6 | 0.6 | 0.6 | 0.6 | 0.6 | 0.6 |
| Li Sparse | 0.1 | 0.1 | 0.1 | 0.1 | 0.1 | 0.1 |
| Maignan scaling | 1 | 1 | 1 | 1 | 1 | 1 |
| Maignan $\nu$ | 0.1 | 0.1 | 0.1 | 0.1 | 0.1 | 0.1 |

**Table 3.** AERONET stations for which the geometries of PARASOL overpasses have been used for forward model comparisons and generic retrieval experiments. Also, aerosol and surface data from these stations have been used to create synthetic measurements for more realistic conditions.

| AERONET Station | Coordinates (lon, lat) |
|---|---|
| Mongu | 23.15, -15.25 |
| Ilorin | 4.34, 8.32 |
| Kanpur | 80.23, 26.51 |
| Banizoumbou | 2.66, 13.54 |
| Beijing | 116.38, 39.98 |



## 2.3 Validation and Comparison Approach for PARASOL Retrievals

We performed retrievals from PARASOL Collection 3 (C3) level-1 data for the year 2008 by both RemoTAP and GRASP and validated the results with AERONET Version 3 Level 2.0 data (http://www.aeronet.gsfc.nasa.gov). Also, we performed a global comparison of retrieved properties. Both RemoTAP and GRASP perform retrievals at the native PARASOL pixel size of 6 km × 6 km. For validation with AERONET, we consider 3X3 pixels centered over all available AERONET sites in 2008. Only comparisons were performed if AERONET, GRASP, and RemoTAP were available. We performed validation for the AOD and

SSA at 550 nm, as well as the Angstrom Exponent (AE) between 440 and 870 nm. Here, it should be noted that the AOD (and hence AE) AERONET product is based om direct sun measurements achieving high AOD accuracy ($\pm$0.01-$\pm$0.02) whereas the SSA is based on inversion of diffuse sky measurements which rely on several retrieval assumptions leading to a moderate accuracy of ($\pm$0.03). Also, inversion level-2 products are strongly filtered on many quality criteria and are only available when AOD(440 nm) > 0.4. This significantly reduces the number of SSA comparisons with AERONET. For the different aerosol

properties we compare the mean retrieved value of the 9 PARASOL pixels corresponding to an overpass to the mean AERONET value within $\pm$1hr of the overpass. We evaluate the differences between the PARASOL retrievals and AERONET against the requirements formulated by the Global Climate Observing System (GCOS). For AOD the GCOS requirement is that the AOD error should be smaller than 0.03 or 10% (whichever is greater). For AERONET validation, this requirement has been modified in the Aerosol-cci study (Popp et al., 2016) to 0.04 or 10% to also take into account the uncertainty in AERONET AOD. For

SSA the GCOS requirement is that the error should be smaller than 0.03. This requirement is not modified for AERONET evaluation given that the 0.03 requirement is considered already loose (Popp et al., 2016). As a validation metric we use the fraction of retrievals with an 'error' (defined as the difference between retrieved value and corresponding AERONET value) smaller than the corresponding requirement. Although for AE no requirement has been formulated, we report the fraction of retrievals with an AE error smaller than 0.2.

For the global comparison between RemoTAP and GRASP products we first grid both products on a $0.1^o$ by $0.1^o$ grid and perform the comparison for all common grid cells. Finally, we compare both RemoTAP and GRASP AOD against a merged MODIS Deep-Blue (DB) (Hsu et al., 2013) and Dark-Target (DT) (Levy et al., 2013) product and the AE againts MODIS-DT (land and ocean). MODIS Collection 6.1 aerosol products (MYD04L2) from the DT and DB algorithms were acquired from the AERIS/ICARE Data and Services Center (http://www.icare.univ-lille.fr), where the unchanged NASA MODIS data are

redistributed.

## 3 Comparison for Simulated Observations

### 3.1 Forward Model Comparison

Figure 1 shows the forward model comparisons between RemoTAP and GRASP. The bias and standard deviation of the differences in radiance are mostly below 2% and for DoLP mostly below 0.005. There is a dependence on wavelength of the

differences: For radiance the differences are larger at 490 and 565 nm, and for DoLP the differences are larger at 670 nm (up



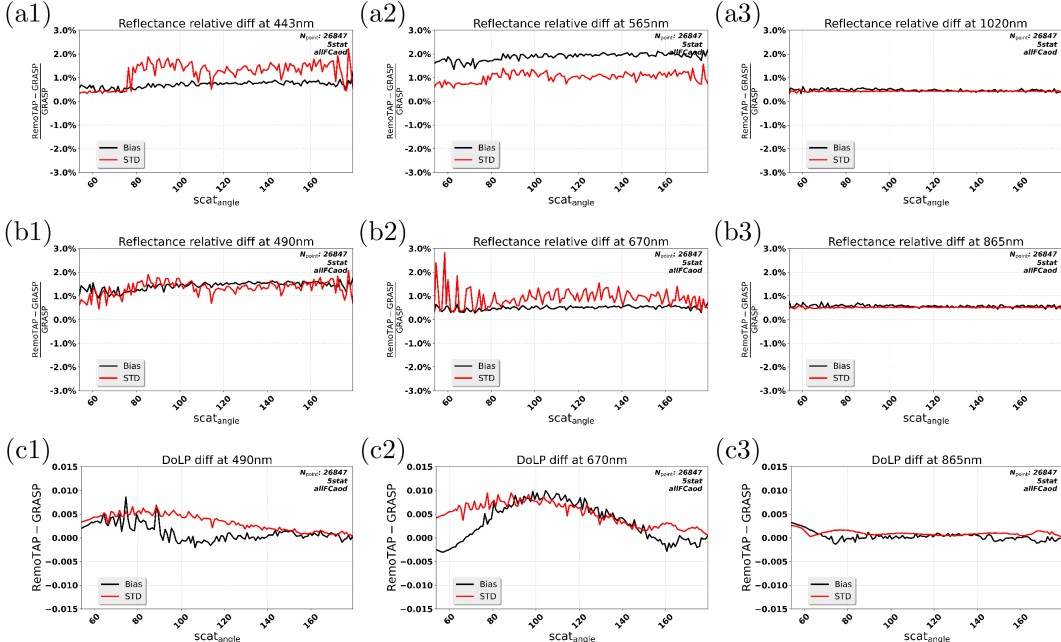

**Figure 1.** Forward model comparisons between RemoTAP and GRASP for the aerosol and surface properties of Table 1 and 2, and geometries of PARASOL overpasses for the AERONET stations of Table 3. Shown are the bias and standard deviation of the differences.

to 0.01 for some angles). The wavelength dependence of the differences can most likely be explained by differences in the way optical aerosol properties are computed from micro-physical properties using the tabulated Mie / T-Matrics / geometric optics calculations, because GRASP and RemoTAP use different methods to interpolate in wavelength and particle size. Overall, the comparison between the GRASP and RemoTAP forward models look reasonably good, and the differences are in general

smaller than the measurement uncertainty for PARASOL. Below we will investigate the importance of the forward model differences by performing RemoTAP retrievals on synthetic measurements created by the GRASP forward model and vice-versa.

## 3.2 Synthetic Retrievals

In reality, the assumed aerosol description in the retrieval will typically be different from the real aerosol properties. Therefore,

it is important to investigate the performance of the algorithms for synthetic measurements created with a more extended and realistic set of aerosol properties than assumed in the retrieval. At the same time, it is important to evaluate the effect of differences due to differences in the forward models of GRASP and RemoTAP. Therefore, the 'RemoTAP-ECHAM' measurements have been created with the RemoTAP forward model and the 'GRASP-AERONET' measurements have been created with the GRASP forward model (see above).





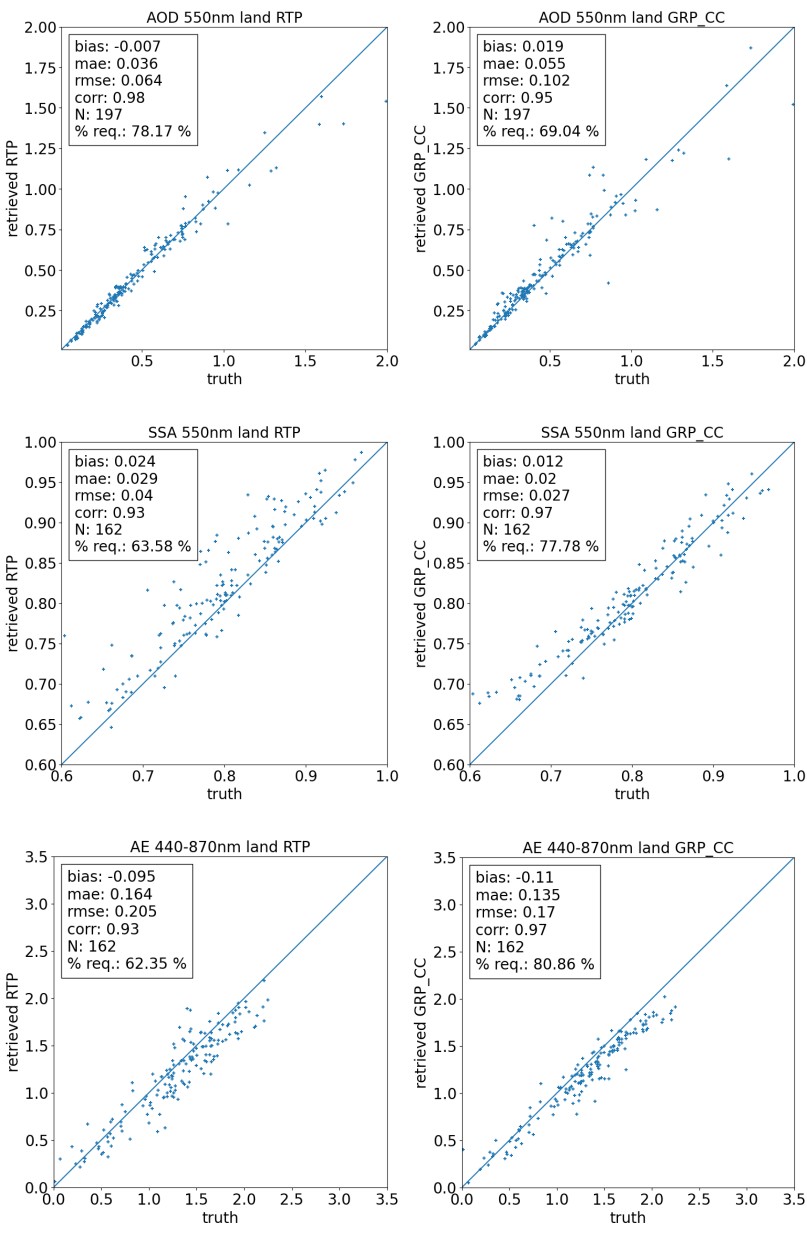

**Figure 2.** Synthetic retrieval results for the 'RemoTAP-ECHAM' synthetic measurements created by the RemoTAP forward model. Shown are the results for RemoTAP (left) and GRASP (right). Properties shown (from top to bottom) are the AOD, SSA, AE. AOD results have been filtered for RemoTAP chi2 < 1 and GRASP minimum residual< 3%. (keeping 197 out of 202 retrievals). AE and SSA results have been additionally filtered for AOD > 0.2.



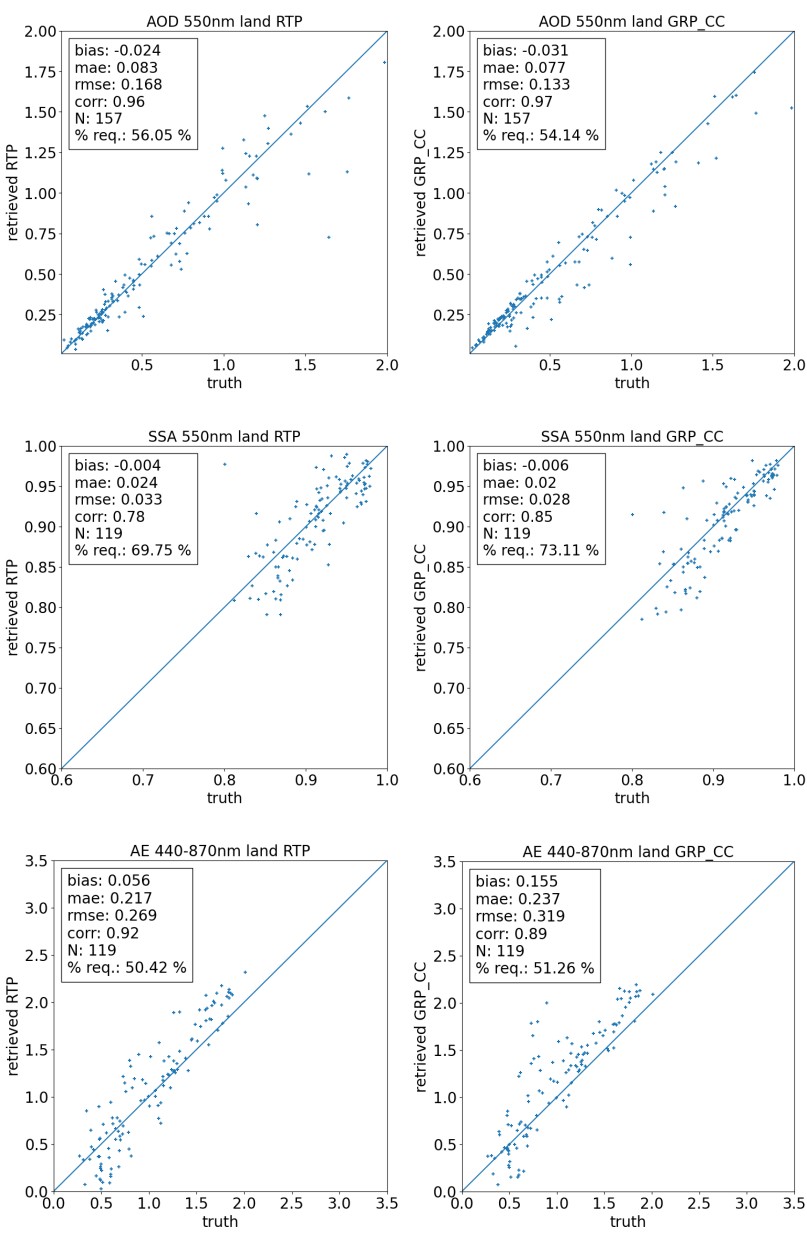

**Figure 3.** Synthetic retrieval results for the 'GRASP-AERONET' synthetic measurements created by the RemoTAP forward model. Shown are the results for RemoTAP (left) and GRASP (right). Properties shown (from top to bottom) are the AOD, SSA, AE. Results have been filtered for RemoTAP chi2 < 1 and GRASP minimum residual< 3% (keeping 157 out of 202 retrievals). AE and SSA results have been additionally filtered for AOD > 0.2.





Figure 2 shows the results for the 'RemoTAP-ECHAM' synthetic measurements created by the RemoTAP forward model. For AOD, GRASP retrievals have a Root Mean Square Error (RMSE) / Mean Absolute Error (MAE) of 0.10 / 0.055 and 69% within the GCOS/CCI requirement, whereas RemoTAP has smaller RMSE / MAE of 0.064 / 0.036 and a larger fraction (78.2%) within the GCOS requirement. For SSA, GRASP shows the best performance with an RMSE/MAE of 0.027/0.024 (77.8% within GCOS/CCI) while RemoTAP has an RMSE/MAE of 0.04/0.029 (63.6% within GCOS). For AE, GRASP also

has a smaller RMSE/MAE (0.17/0.135) than RemoTAP (0.21/0.164). Figure 3 shows the results for the 'GRASP-AERONET' synthetic measurements created by the GRASP forward model. For AOD, GRASP has an RMSE/MAE of 0.13/0.077 and 54.1% within GCOS/CCI requirements, whereas RemoTAP has an RMSE/MAE of 0.17/0.083 and 56.1% within GCOS/CCI requirements. For SSA, GRASP has an RMSE/MAE of 0.028 /0.02 with 73.1% retrievals within GCOS/CCI requirements and a bias of -0.006. RemoTAP has an RMSE/MAE of 0.033/0.024 and a fraction of retrievals within GCOS/CCI requirements

(69.8%) but a similar bias (-0.004). For AE RemoTAP has a smaller RMSE and MAE on the 'GRASP-AERONET' synthetic measurements (RMSE/MAE of 0.269/0.217 vs 0.319/0.237). To conclude, we see that for AOD the best performance is obtained for the algorithm which forward model has been used to generate the synthetic measurements. For SSA, GRASP has smaller RMSE/MAE for both sets of synthetic measurements. For AE, interestingly GRASP performs has smaller RMSE/-MAE for the SRON-ECHAM synthetic measurements and RemoTAP has smaller RMSE/MAE for the GRASP-AERONET

synthetic measurements. However, overall the difference in performance on synthetic measurements are small and both Remo-TAP and GRASP have demonstrated robustness against aerosol assumptions and forward model differences and are capable of performing accurate retrievals under different circumstances.

## 4 Comparison for real PARASOL observations

### 4.1 Validation with AERONET

#### 4.1.1 Retrievals over Land

Figure 4 shows the validation of $AOD_{550nm}$, AE, and $SSA_{550nm}$ against AERONET for both RemoTAP and GRASP retrievals over land. Metrics for AOD validation for different AOD ranges can be found in Table 5, and metrics for SSA, AE, and fine- and coarse- mode AOD in Table 6. Only common pixels are included in this validation. For comparison, we also show validation results of the previous versions of the RemoTAP (Lacagnina et al., 2017) and GRASP (Chen et al., 2020) in Table 4. Although

these validation results correspond to a different year (2006) and a different spatial gridding ($0.2^o$), we can clearly see that the latest versions of RemoTAP and GRASP have substantially improved compared to the earlier version(s). The latest algorithm versions used in the present study show closer agreement with AERONET for AOD and SSA. For AE, the present version of GRASP (CC) shows similar agreement with AERONET as the previous (HP) version but RemoTAP improved significantly, removing a large positive bias at small AOD, also reported by Tsikerdekis et al. (2023). If we compare the performance between

RemoTAP and GRASP, we see that for AOD both algorithms show very similar performance against AERONET, both in terms of MAE/RMSE and bias but also for the fraction of the retrievals within GCOS/CCI requirements. For AE, GRASP has still





**Table 4.** Validation metrics of the previous RemoTAP (Lacagnina et al., 2017) and GRASP (Chen et al., 2020) PARASOL products. Note that for SSA the validation was not separated in retrievals over land and ocean because not enough data points are available for comparison. The validation has been performed for the year 2006 and a spatial gridding of $0.2^o$ because the RemoTAP global product is only available for this year and spatial gridding. The difference in gridding explains the different values for RMSE than found by Chen et al. (2020).

|  | RMSE (land / ocean) | %GCOS (land / ocean) | Bias (land / ocean) | Bias AOD <0.2 (land / ocean) |
|---|---|---|---|---|
| AOD (550nm) |  |  |  |  |
| RemoTAP (V2017) | 0.181 / 0.122 | 32.8 / 46.1 | 0.02 / 0.04 | 0.07 / 0.03 |
| GRASP-HP (V2020) | 0.164 / 0.127 | 34.3 / 32.4 | 0.05 / 0.07 | 0.06 / 0.06 |
| GRASP-M (V2020) | 0.165 / 0.068 | 42.8 / 63.3 | 0.0 / 0.01 | 0.0 / 0.01 |
| AE (440-870nm) |  |  |  |  |
| RemoTAP (V2017) | 0.63 / 0.27 | n/a | 0.05 / 0.06 | n/a |
| GRASP-HP (V2020) | 0.382 / 0.39 | n/a | -0.16 / -0.22 | n/a |
| GRASP-M (V2020) | 0.53 / 0.365 | n/a | 0.1 / 0.02 | n/a |
| SSA 550nm (land+ocean) |  |  |  |  |
| SRON/RTP | 0.040 | 44.8 | 0 | n/a |
| GRASP-HP | 0.056 | 40.7 | -0.03 | n/a |
| GRASP-M | 0.061 | 36.9 | -0.03 | n/a |

a smaller RMSE because RemoTAP still shows a small overestimate at low AE values and an underestimate at large values, although the difference in RMSE with GRASP is very small. The comparison of fine- and coarse mode AOD confirms the better performance of GRASP for aerosol size retrieval. For SSA, RemoTAP shows a s smaller bias against AERONET then GRASP. The RMSE for SSA comparable for both RemoTAP and GRASP and the fraction of retrievals within the GCOS/CCI requirements is 61.4 for RemoTAP and 55.2% for GRASP.

**Table 5.** Comparison of RemoTAP and GRASP AOD550nm over land.

| AOD range | RemoTAP AOD550nm | | | | GRASP AOD550nm | | | |
|---|---|---|---|---|---|---|---|---|
|  | RMSE | MAE | BIAS | GCOS(%) | RMSE | MAE | BIAS | GCOS(%) |
| Full | 0.10 | 0.063 | -0.009 | 54.2 | 0.10 | 0.062 | -0.002 | 53.6 |
| [0-0.2] | 0.065 | 0.041 | 0.009 | 65.8 | 0.072 | 0.044 | 0.013 | 63.7 |
| [0.2-0.7] | 0.13 | 0.092 | -0.024 | 33.7 | 0.11 | 0.082 | -0.018 | 36.3 |
| [0.7-4.2] | 0.242 | 0.175 | -0.127 | 36.0 | 0.233 | 0.166 | -0.095 | 36.0 |

Figures 5 and 6 show the RMSE and bias in AOD, AE, and SSA, respectively for different regions / countries: Europe (EU), Asia (AS), Africa (AF), Oceania (OC), South America (SA), China (CN), and the Sahara (Sah). In AOD, we see many similarities between RemoTAP and GRASP in the regional dependence of RMSE, with large RMSE (in absolute sense) over Asia, China, and the Sahara for both algorithms, which are regions with large AOD. Low values for RMSE are found over Europe,

8000




**Figure 4.** Validation of RemoTAP (top panels) and GRASP (bottom panels) retrievals over land with AERONET, for AOD (left), AE (middle), and SSA (right) at 550 nm.

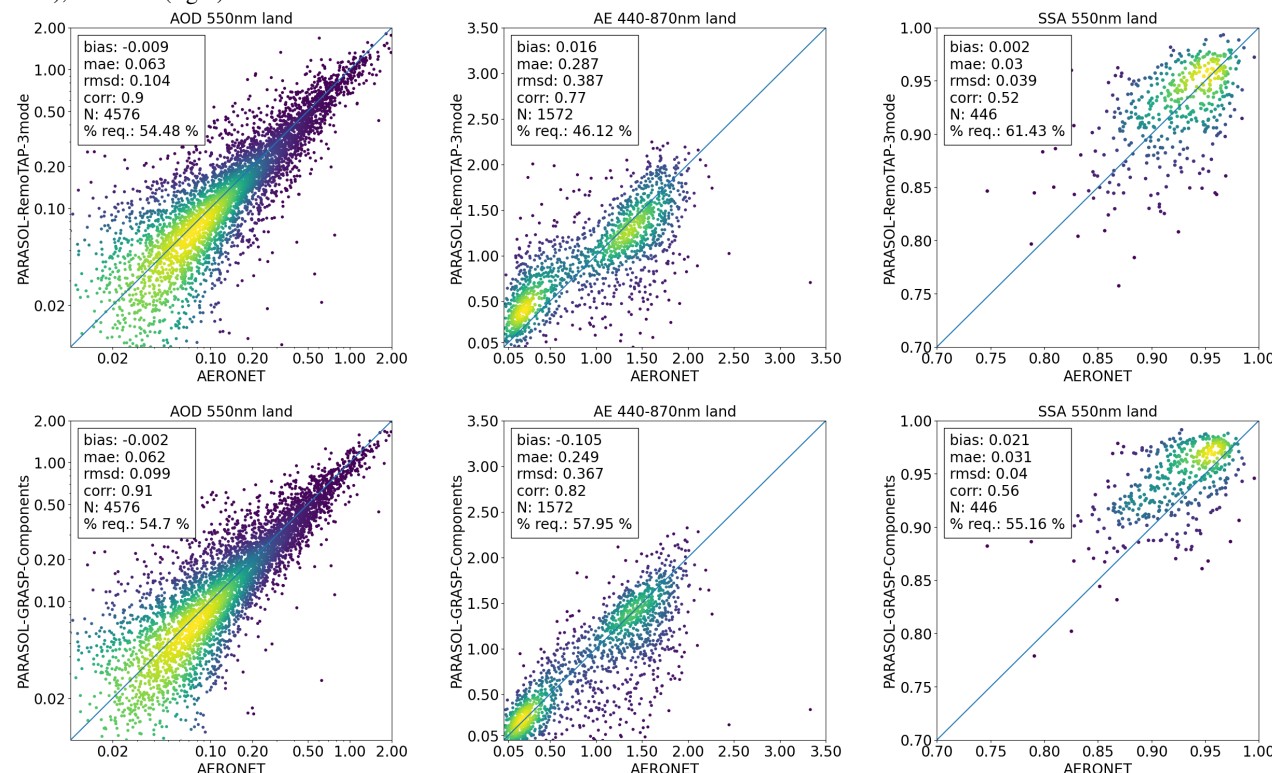

**Table 6.** Comparison of RemoTAP and GRASP for SSA, AE, and fine- and coarse mode AOD over land.

| Property | RemoTAP | | | | GRASP | | | |
|---|---|---|---|---|---|---|---|---|
| | RMSE | MAE | BIAS | GCOS(%) | RMSE | MAE | BIAS | GCOS(%) |
| SSA | 0.039 | 0.03 | 0.002 | 61.4 | 0.04 | 0.031 | 0.021 | 55.1 |
| AE | 0.39 | 0.29 | 0.016 | 46.1 | 0.37 | 0.25 | -0.11 | 58.0 |
| $AOD_{fine}$ | 0.085 | 0.050 | -0.004 | 61.5 | 0.071 | 0.044 | -0.006 | 65.4 |
| $AOD_{coarse}$ | 0.082 | 0.049 | -0.009 | 63.8 | 0.074 | 0.043 | -0.003 | 68.9 |

and South- and North America. Over Oceania, RemoTAP has larger RMSE than GRASP (especially at smaller wavelengths). Looking at the bias, RemoTAP has smaller bias over Europe, Asia, and China, and a larger bias over Oceania, and the Sahara. Both algorithms have similar bias over Africa (although opposite in sign), North Ameria, South America, and the USA. The regional dependence in RMSE and bias for AE shows a similar pattern as for AOD for both RemoTAP and GRASP, with the

exception that GRASP, like RemoTAP, also shows large RMSE and bias over Oceania, while for AOD this region was only problematic for RemoTAP. Further, we see from Fig. 5 and 6 that overall the AE behaves similar for different wavelength pairs,





although the pair 550-870 nm shows smaller RMSE and bias over Oceania. For SSA we see that GRASP shows larger variation in RMSE over regions than RemoTAP. For bias, GRASP shows also larger variation over region than RemoTAP but there is a clear correlation between the 2 algorithms. We also see that both RemoTAP and GRASP show worse performance for SSA at 290 870 nm than the other wavelengths, which is most likely caused by the fact that AOD at 870 nm is typically small.

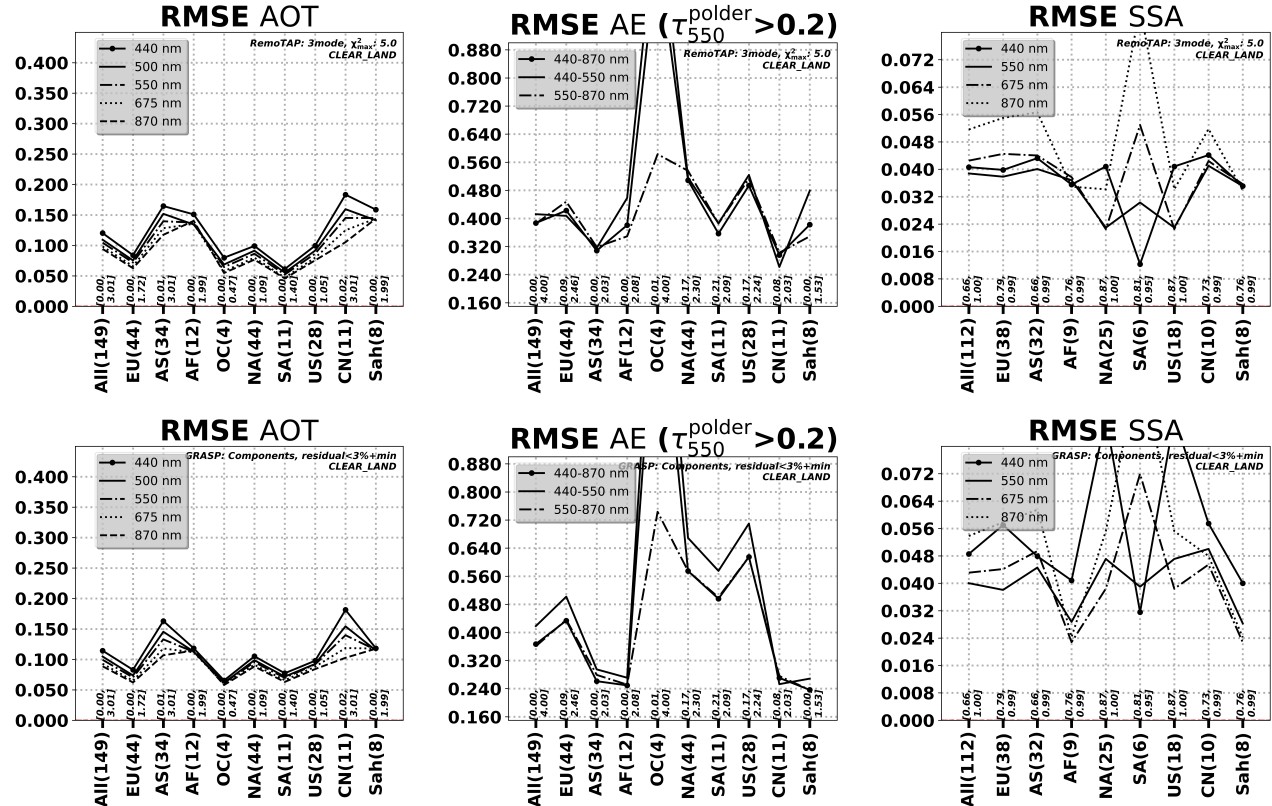

**Figure 5.** RMSE for different regions/countries for AOD (left), AE (middle), and SSA (right). Upper panels correspond to RemoTAP retrievals and lower panels to GRASP retrievals. Metrics are shown for all stations (all), Europe (EU), Asia (AS), Africa (AF), Oceania (OC), South America (SA), China (CN), and the Sahara (Sah).

### 4.1.2 Retrievals over Ocean

Figure 7 shows the validation of $AOD_{550nm}$, AE, and $SSA_{550nm}$ against AERONET for both RemoTAP and GRASP retrievals (common pixels) over ocean. Metrics for AOD validation for different AOD ranges can be found in Table 7, and metrics for SSA, AE, and fine- and coarse- mode AOD in Table 8. As expected, the performance for AOD and AE of both algorithms 295 over ocean is better than over land and both algorithms perform very well against AERONET. We also note that GRASP has a larger GCOS-fraction and smaller MAE, RMSE, and bias for AOD than RemoTAP although the differences are small ($\leq 0.01$). Zooming in to different AOD ranges (Table 7), we see the same general behavior. For AE, both RemoTAP and GRASP show a





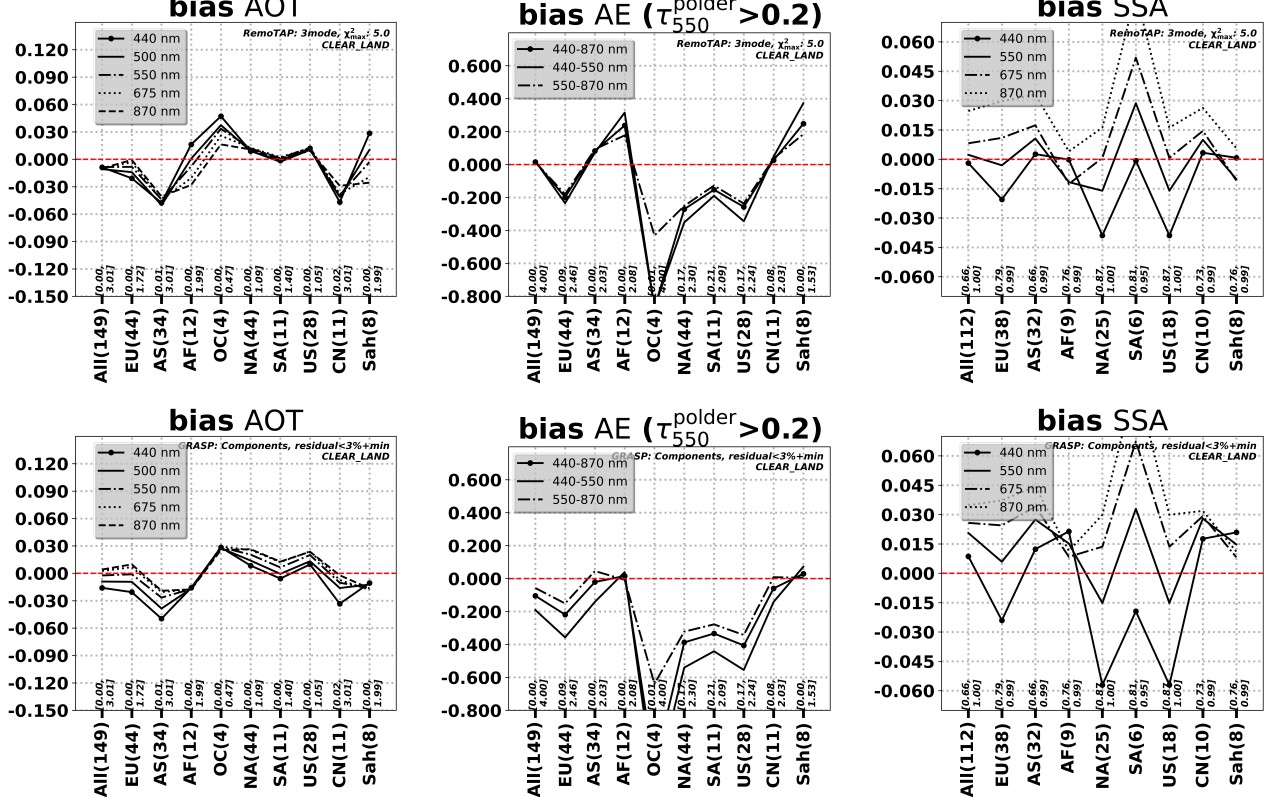

**Figure 6.** Bias for different regions for AOD (left), AE (middle), and SSA (right). Upper panels correspond to RemoTAP retrievals and lower panels to GRASP retrievals.

tendency to underestimate AE for larger values. Like for AOD, GRASP shows smaller MAE, RMSE and bias than RemoTAP. For SSA, the agreement with AERONET is similar for RemoTAP and GRASP but the number of co-locations is too small to

draw further conclusions. Comparing the validation results over ocean to those of the previous versions of the RemoTAP and GRASP products in Table 4, we can see that the RMSE of the RemoTAP AOD improved by about a factor 2 and the GRASP product improved by even more compared to the previous GRASP-HP product and by about 30% compared to the GRASP-M product. Also the bias of RemoTAP improved (overall and for AOD smaller than 0.2) and the bias of GRASP improved compared to the previous GRASP-HP product. For AE, GRASP improved considerably while the RemoTAP performance for

AE stays similar as for the previous product version. Like over land, GRASP has smaller RMSE for AE over ocean and this is consistent with the results for fine- and coarse mode AOD.





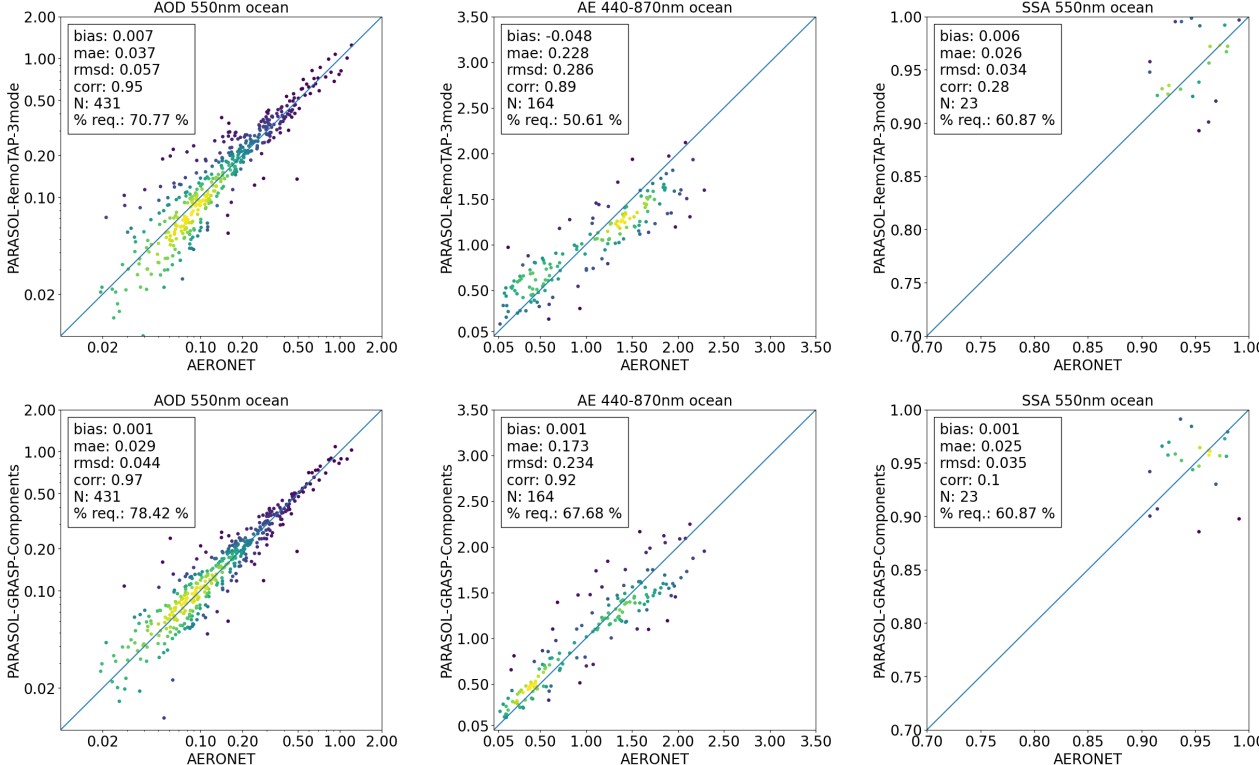

**Figure 7.** Validation of RemoTAP (top panels) and GRASP (bottom panels) retrievals with AERONET, for AOD (left), AE (middle), and SSA (right) at 550 nm.

**Table 7.** Comparison of RemoTAP and GRASP AOD550nm over ocean

| AOD range | RemoTAP AOD550nm | | | | GRASP AOD550nm | | | |
|---|---|---|---|---|---|---|---|---|
| | RMSE | MAE | BIAS | GCOS(%) | RMSE | MAE | BIAS | GCOS(%) |
| Full | 0.057 | 0.037 | 0.008 | 70.9 | 0.044 | 0.030 | 0.002 | 77.2 |
| [0-0.2] | 0.047 | 0.030 | 0.011 | 76.9 | 0.033 | 0.023 | 0.008 | 83.8 |
| [0.2-0.7] | 0.062 | 0.044 | 0.007 | 61.7 | 0.051 | 0.035 | -0.005 | 67.2 |
| [0.7-4.2] | 0.15 | 0.136 | -0.04 | 27.3 | 0.131 | 0.114 | -0.049 | 36.36 |

## 4.2 Comparison for Global Data Products

We performed global processing with RemoTAP and GRASP and compared the gridded data products ($0.1 \times 0.1^o$). Both RemoTAP and GRASP provide a similar number of valid retrievals after data filtering. Also, we perform a comparison with
MODIS data products.





**Table 8.** Comparison of RemoTAP and GRASP for SSA, AE, and fine- and coarse mode AOD over ocean.

| Property | RemoTAP | | | | GRASP | | | |
|---|---|---|---|---|---|---|---|---|
| | RMSE | MAE | BIAS | GCOS(%) | RMSE | MAE | BIAS | GCOS(%) |
| SSA | 0.034 | 0.026 | 0.006 | 60.9 | 0.035 | 0.025 | 0.001 | 60.9 |
| AE | 0.29 | 0.23 | -0.048 | 50.6 | 0.23 | 0.17 | 0.001 | 67.7 |
| $AOD_{fine}$ | 0.090 | 0.050 | 0.021 | 67.7 | 0.057 | 0.041 | 0.022 | 61.4 |
| $AOD_{coarse}$ | 0.070 | 0.038 | -0.014 | 74.6 | 0.036 | 0.026 | -0.018 | 81.8 |

### 4.2.1 Comparison of AOD

Figure 8 shows maps of the mean AOD for GRASP and RemoTAP, respectively, and a map of the mean differences. Overall, GRASP and RemoTAP show the same AOD pattern with high AOD over the Sahara (Dust) and equatorial Africa (biomass burning) and the outflow of dust and biomass burning aerosol over the Atlantic ocean. Also, high AOD values are retrieved

over polluted areas in east Asia, and the Ganges valley. Also interesting is the high mean AOD over Siberia, which is related to boreal forest fires. If we look at the differences between RemoTAP and GRASP, for most of the globe the differences are small. Exceptions are the Sahara/Arabia where RemoTAP AOD is higher by 0.05-0.10 and equatorial Africa where RemoTAP AOD is lower than GRASP by almost 0.15. Also notable is the difference (especially in relative sense) over the most southern part of the ocean where RemoTAP retrieves lower AOD than GRASP.

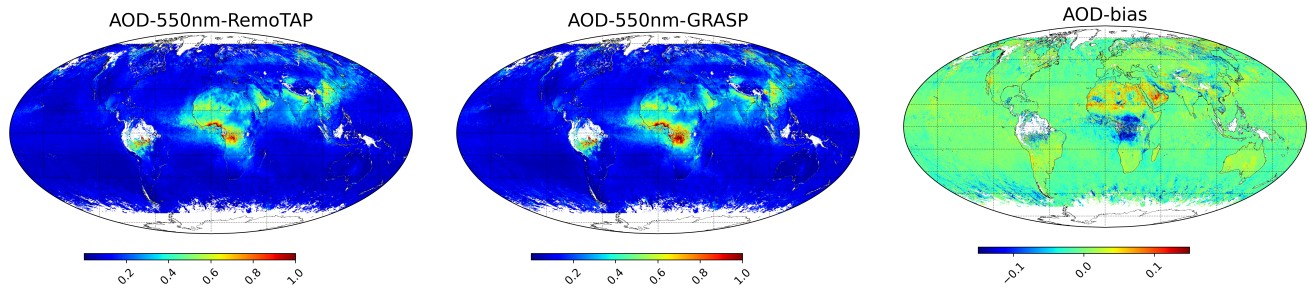

**Figure 8.** Annual mean AOD for (left) RemoTAP, (middle) GRASP, and (right) the bias between them (RemoTAP-GRASP).

Figure 9 shows scatter plots of the RemoTAP AOD versus the GRASP AOD at 550nm, separately for retrievals over land and ocean. Also, a histogram of the differences is shown. Over land, the RMSD is 0.12 and the difference in the mean (RemoTAP-GRASP) is 0.01. The RMSD and bias are larger at 443 nm (0.14 and 0.026, respectively) and comparable at 865 nm (not shown). Over ocean, the RMSD is 0.038 nm and the bias is -0.008. The differences are larger at 443 nm and smaller differences at 865 nm (not shown).





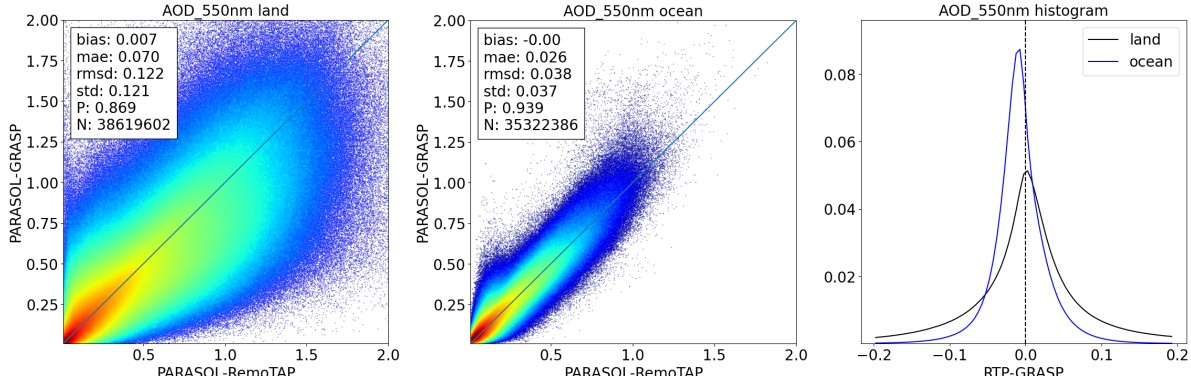

**Figure 9.** AOD (550 nm) scatter plots (GRASP versus RemoTAP) for retrievals over land (left) and ocean (middle) and a histogram of the differences (right), where the blue line corresponds to retrievals over ocean and the black line to retrievals over land.

### 4.2.2    Comparison of AE, Fine- and Coarse Mode AOD

Figure 10 shows maps of the mean AE (440-865 nm) for GRASP and RemoTAP, respectively, and the map of the mean differences. These maps include only retrievals for cases where $AOD_{550nm} > 0.2$. The AE maps for RemoTAP and GRASP both show a very similar overall pattern. Small AE values (larger sizes) occur over desert and over the open ocean and correspond mostly to situations dominated by Dust or Sea Salt. Larger AE values (smaller sizes) occur over areas with anthropogenic pollution (Asia, Europe, North- and South America) and over areas with biomass burning (southern Africa, South America, Indonesia, Australia). Most important differences between RemoTAP and GRASP arise over the Sahara and Middle East, where RemoTAP retrieves higher AE than GRASP, which is consistent with the AERONET comparison showing overestimated AE values for big particles provided by RemoTAP. On the other hand, the maximum differences (where RemoTAP can be 0.5 higher than GRASP) are not expected from the AERONET comparison.

Figure 11 shows scatter plots of the RemoTAP AE versus the GRASP AE, separately for retrievals over land and ocean. Only retrievals with AOD>0.2 are included. Also, a histogram of the differences is shown. Over land, the RMSD is 0.33 while over ocean the RMSD is 0.23. Over land, the agreement is very good for larger values of AE (small particle size) and a bit worse for lower AE values (large particles size), where RemoTAP retrieves larger AE. Over ocean, the agreement is very good for both low and high AE values.

When using a lower AOD threshold of 0.1 (not shown) the agreement gets worse with RMSD values of 0.48 and 0.36 over land and ocean, respectively. Over land, the overall scatter of the data increases compared to the higher AOD threshold, while over ocean also we see some specific cases where RemoTAP retrieves small AE (close to 0) and GRASP retrieves values up to 1.5. To investigate the dependence of AE difference on AOD in more detail, Figure 12 shows the AE difference as a function of AOD. We can see the AE difference depends strongly on AOD. Over land, there is a large positive bias ( 0.4) at AOD=0.10 which decreases gradually to 0 at AOD=0.4. Over ocean, there is a bias of -0.45 at AOD=0.05 which decreases more rapidly





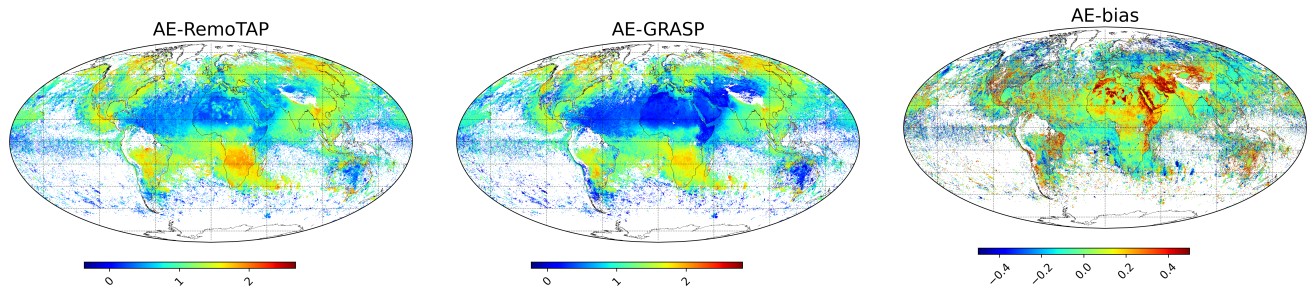

**Figure 10.** Annual mean AE for (left) RemoTAP, (middle) GRASP, and (right) the bias between them (RemoTAP-GRASP). Only retrievals with $AOD_{550nm} > 0.2$ are included.

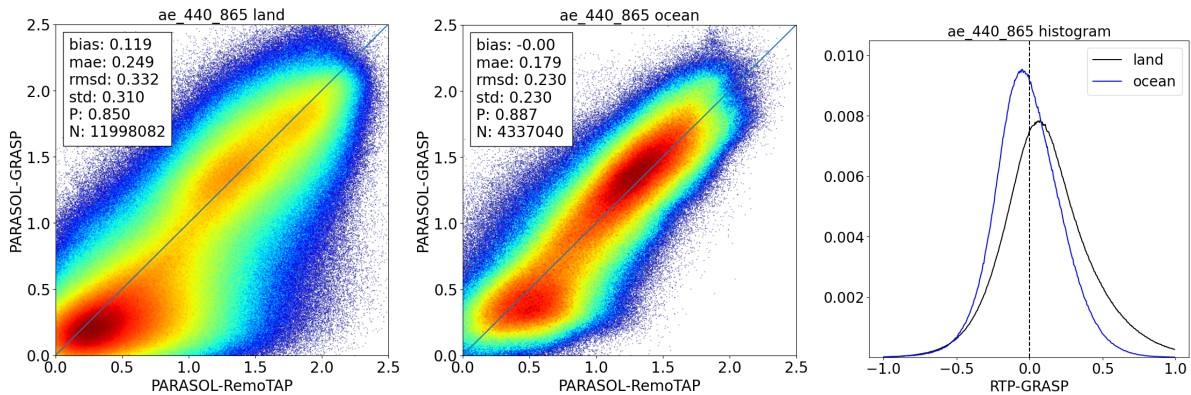

**Figure 11.** AE scatter plots (GRASP versus RemoTAP) for retrievals over land (left) and ocean (middle) and a histogram of the differences (right), where the blue line corresponds to retrievals over ocean and the black line to retrievals over land. Only retrievals with $AOD_{550nm} > 0.2$ are included.

with AOD than over land. The strong dependence of AE difference on AOD results from the fact that that at small AOD, the AE calculation is very sensitive to even small errors in AOD at the different wavelengths.

Figure 13 shows the comparison for the fine mode AOD and Figure 14 for the coarse mode AOD. Over land, the RMSD for the fine mode AOD is smaller than for the coarse mode AOD (RMSD 0.063 vs 0.107). This means that for the total AOD, most of the differences can be explained by differences in the coarse mode AOD. Further, we see that for the fine mode AOD RemoTAP retrieves smaller values than GRASP towards higher AOD while for the coarse mode the opposite is observed. For retrievals over ocean, the RMSD for the coarse mode AOD is smaller than for the fine mode. Further, for the fine mode RemoTAP is systematically smaller than GRASP (-0.02 bias) while for the coarse mode RemoTAP is larger (0.016 bias), such that the total AOD bias between RemoTAP and GRASP is very small.



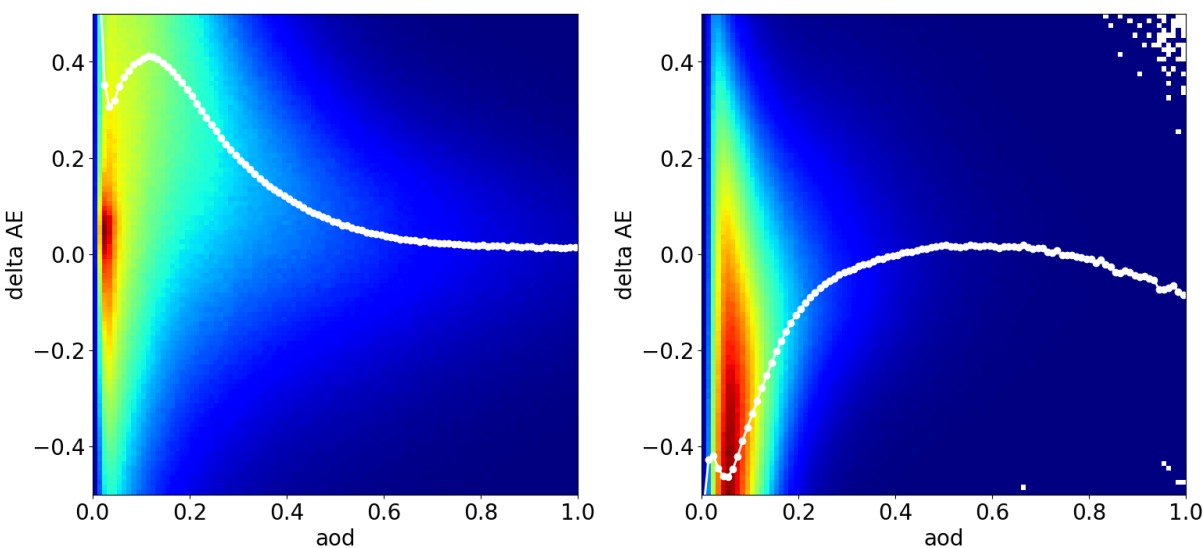

**Figure 12.** Difference in AE (RemoTAP-GRASP) versus AOD for retrievals over land (left) and ocean (right)

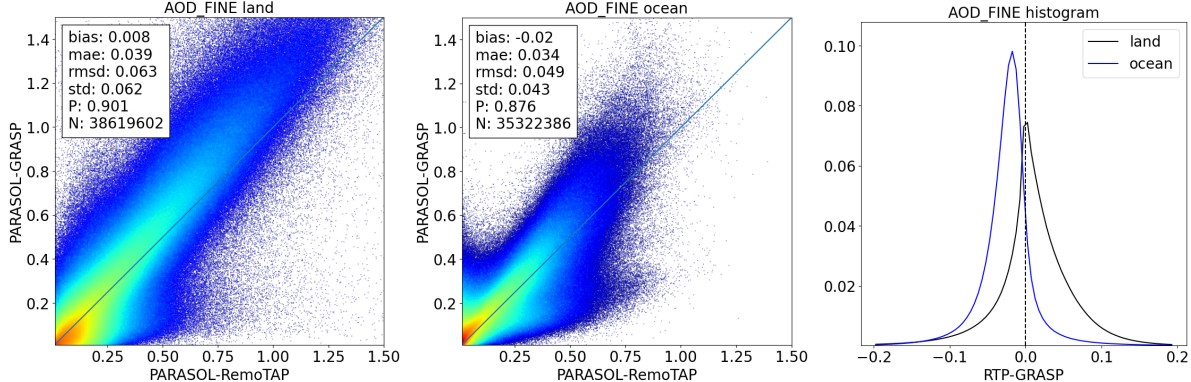

**Figure 13.** Fine mode AOD (550 nm) scatter plots (GRASP versus RemoTAP) for retrievals over land (left) and ocean (middle) and a histogram of the differences (right), where the blue line corresponds to retrievals over ocean and the black line to retrievals over land.





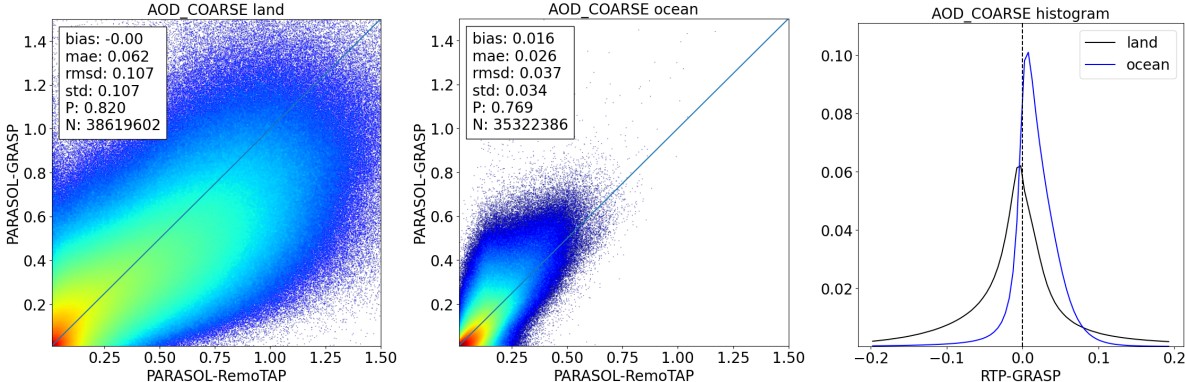

**Figure 14.** Coarse mode AOD (550 nm) scatter plots (GRASP versus RemoTAP) for retrievals over land (left) and ocean (middle) and a histogram of the differences (right), where the blue line corresponds to retrievals over ocean and the black line to retrievals over land.

### 4.2.3 Comparison of SSA and AAOD

Figure 15 shows maps of the mean SSA (550 nm) for GRASP and RemoTAP, and the bias. Overall, the maps show similar patters but there are also important differences. In general, RemoTAP retrieves higher SSA over ocean and lower SSA over land. Larger differences occur over equatorial Africa (biomass burning region) where RemoTAP retrieves significantly higher (> 0.10 difference) SSA than GRASP, over ocean but also over land (in contrast to other land regions). The difference is especially apparent over ocean. Also, over India and the Indian ocean there are notable differences in SSA between RemoTAP and GRASP. Here, over land RemoTAP retrieves a mean SSA of 0.90 while GRASP retrieves 0.95. Over ocean, the difference is opposite: RemoTAP retrieves SSA 0.95-1 while GRASP retrieved 0.90. Both RemoTAP and GRASP show an unexpected sharp transition between land and ocean in this region. A similar pattern can be seen at the west coast of the USA.

Figure 16 shows scatter plots of the RemoTAP SSA versus the GRASP SSA at 550 nm, separately for retrievals over land and ocean. Also, a histogram of the differences is shown. Over land, the RMSD is 0.043 whereas the overall bias is < 0.01. There is some compensation between different areas, as is apparent from the world map, but overall, the agreement can be considered good over land. Over ocean the differences are substantially larger, with an RMSD and bias of 0.074 and 0.053, respectively. Here, RemoTAP retrieves higher SSA than GRASP, as was already seen from the world map. As noted above, the AERONET SSA validation over ocean does not have sufficient points to indicate whether the difference is caused by errors in RemoTAP or GRASP. Clearly, there is a need for more SSA validation points over ocean. Figure 16 also shows the comparison for AAOD. Over land, the overall bias is relatively small (0.003) but there is considerable scatter between RemoTAP and GRASP. Over ocean, there is a clear bias between RemoTAP and GRASP where GRASP retrieves higher AAOD than RemoTAP. This is expected because GRASP retrieves smaller SSA than RemoTAP over ocean and comparable AOD, which should result in a higher AAOD.





Figure 17 shows the SSA difference (RemoTAP-GRASP) as a function of AOD (mean of RemoTAP and GRASP) for retrievals over land and ocean, respectively. Over land, we see that for AOD < 0.15 the difference between RemoTAP and GRASP is largest where the mean difference reaches -0.10 for the lowest AOD values. The SSA difference decreases (in absolute sense) rapidly till AOD = 0.15 where the SSA difference is -0.01 and slowly decreases further to higher AOD. Despite the fact that on average the RemoTAP SSA is lower than the GRASP SSA, we see that for a substantial number of

retrievals the GRASP SSA is higher. These retrievals correspond mostly to biomass burning retrievals (see above).

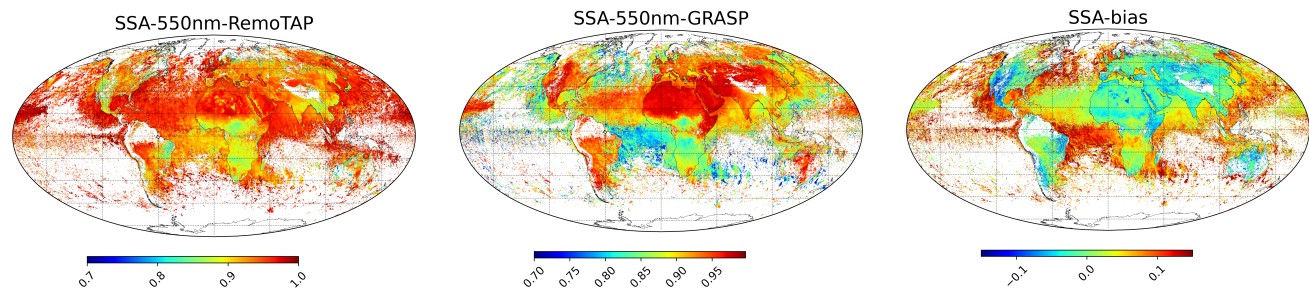

**Figure 15.** Annual mean SSA for (left) RemoTAP, (middle) GRASP, and (right) the bias between them (RemoTAP-GRASP). Only retrievals with $AOD_{550nm}$ are included.

### 4.3    Summary of Global Aerosol Comparison

| Property | Surface | New Products | | Old Products (GRASP-HP) | | Old Products (GRASP-M) | |
|---|---|---|---|---|---|---|---|
| | | RMSD | bias | RMSD | bias | RMSD | bias |
| AOD (550 nm) | Land | 0.12 | 0.007 | 0.22 | 0.01 | 0.234 | 0.08 |
| SSA (550 nm) | Land | 0.043 | -0.01 | 0.083 | 0.06 | 0.077 | 0.04 |
| AE (440-865 nm) | Land | 0.33 | 0.12 | 0.58 | 0.33 | 0.58 | 0.24 |
| AOD (550 nm) | Ocean | 0.038 | 0.00 | 0.073 | -0.02 | 0.078 | 0.04 |
| SSA (550 nm) | Ocean | 0.074 | 0.053 | 0.066 | 0.02 | 0.079 | 0.05 |
| AE (440-865 nm) | Ocean | 0.23 | 0.00 | 0.22 | 0.01 | 0.31 | 0.19 |

**Table 9.** RMSD and bias between GRASP and RemoTAP for global retrievals for the year 2008 (Jan-Nov) for the properties AOD, SSA, and AE. The table shows both the comparision of the new products and the previous versions of the RemoTAP (Lacagnina et al., 2017) and GRASP (Chen et al., 2020) products

Table 9 shows a summary of the global comparison between GRASP and RemoTAP for 2008. For reference, also the global comparison of the old RemoTAP and GRASP products is shown (for 2006 and at $0.2^o$ spatial gridding). We can see that for AOD the agreement between GRASP and RemoTAP improved signifcantly, with a reduction in RMSD by about a factor 2.

For AE over land, both the RMSD and bias improved by about a factor 2, while for AE over ocean the agreement stays similar



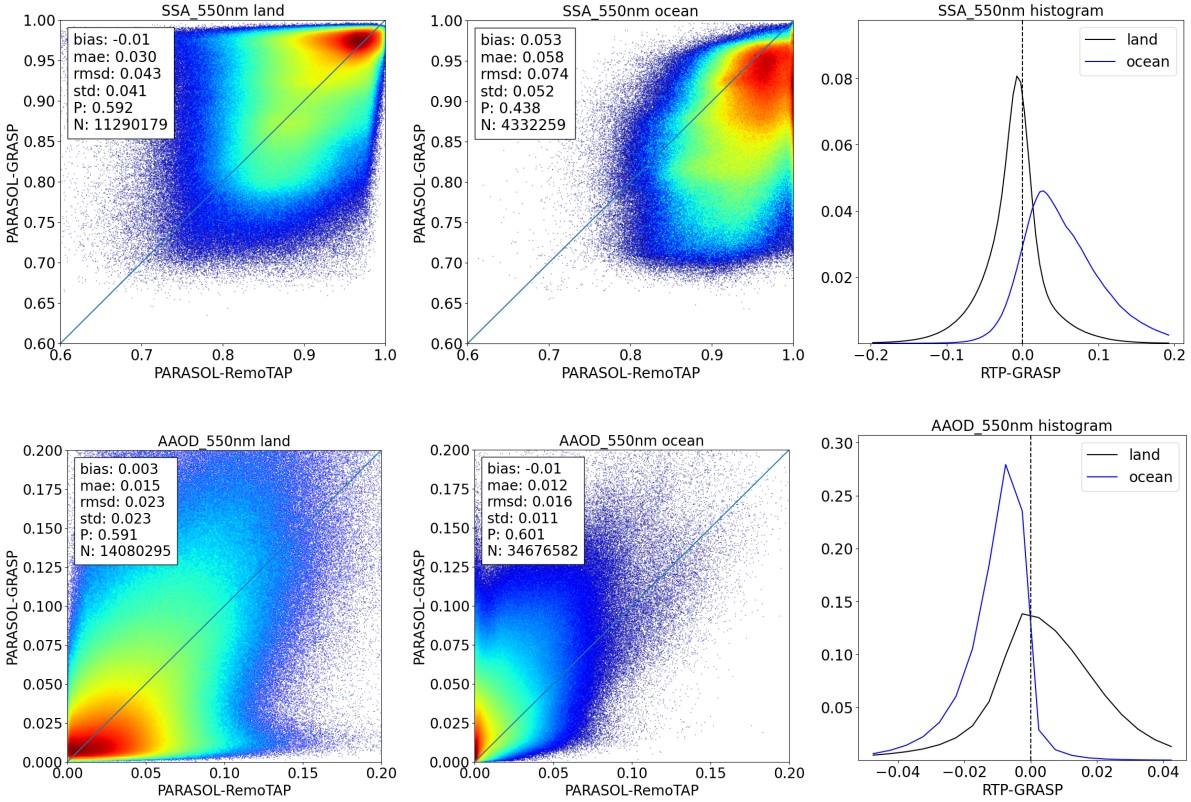

**Figure 16.** SSA and AAOD (550 nm) scatter plots (GRASP versus RemoTAP) for retrievals over land (left) and ocean (middle) and a histogram of the differences (right), where the blue line corresponds to retrievals over ocean and the black line to retrievals over land.

as for the previous product versions. For SSA, the agreement over land is reasonable and significantly improved compared to the previous product versions. Over ocean, the agreement is poor with a bias of 0.053 (RemoTAP retrieves higher SSA than GRASP) and the agreement did not improve compared to the previous product versions.

## 4.4 Surface Properties

Table 10 shows the RMSD and bias for the comparison of the isotropic BRDF parameters (see section B1) between RemoTAP and GRASP at different wavelengths. We can see that RemoTAP retrieves a higher isotropic BRDF than GRASP at all wavelengths, with a bias ranging from 0.018 to 0.028. The RMSD ranges from 0.018 to 0.036. The largest differences (in absolute sense) occur over the Sahara and the Arabian Peninsula (not shown), where the largest difference in AE aerosol parameter is also observed. Only at 865 nm RemoTAP retrieves a smaller isotropic BRDF parameter than GRASP over equitorial Africa.

Table 11 shows the RMSD and bias for the directional BRDF parameters and the BPDF. RemoTAP retrieves larger values for the directional BRDF parameters and substantially lower BPDF scaling parameter than GRASP. Like for all surface parameters, the difference is largest over the Sahara and the Arabian peninsula. To investigate the difference in the BPDF scaling





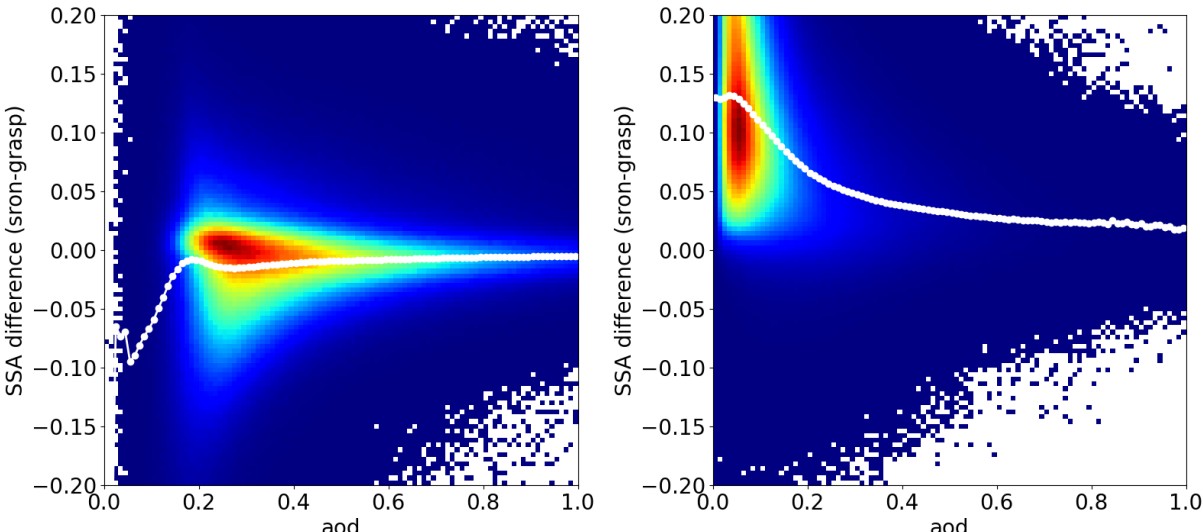

**Figure 17.** Difference in SSA (RemoTAP-GRASP) versus AOD for retrievals over land (left) and ocean (right)

parameter in more detail, Figure 18 shows the difference in BPDF scaling parameter as a function of the difference in isotropic BRDF parameter (490 nm). There is a very clear dependence when the difference in isotropic BRDF parameter is in the range

0-0.05. Here, the difference in BPDF scaling gets more negative with increasing difference in isotropic BRDF parameter. So, a larger isotropic BRDF is compensated with a smaller BPDF scaling in RemoTAP. This can be explained by the fact that the Fresnel reflection matrix that is scaled in the BPDF model also contributes to the BRDF. This would imply that either the surface polarization is too small in RemoTAP or too large in GRASP.

**Table 10.** RMSD and bias for comparison of the isotropic BRDF parameters at different wavelengths

|          | iso-443nm | iso-490nm | iso-565nm | iso-670nm | iso-865nm | iso-1020nm |
|----------|-----------|-----------|-----------|-----------|-----------|------------|
| **RMSD** | 0.019     | 0.018     | 0.024     | 0.031     | 0.034     | 0.036      |
| **BIAS** | 0.018     | 0.022     | 0.027     | 0.028     | 0.023     | 0.023      |

**Table 11.** Statistical metrics for different models

| Metric   | BRDF2 | BRDF3 | BPDF |
|----------|-------|-------|------|
| **RMSD** | 0.143 | 0.042 | 1.68 |
| **BIAS** | 0.051 | 0.013 | -2.3 |

Figure 19 shows the difference in AOD between RemoTAP and GRASP as a function of difference in retrieved isotropic

BRDF parameter (at 490 nm) and the difference in BPDF scaling parameter. It can be seen that for cases where RemoTAP




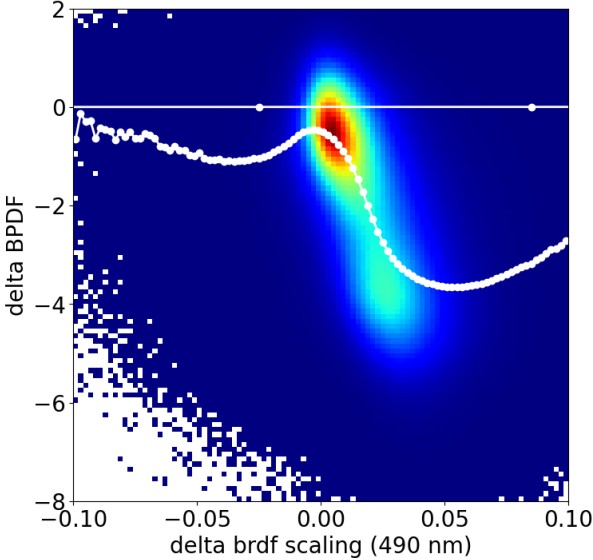

**Figure 18.** Difference in BPDF scaling parameters as a function of the difference in the isotropic BRDF parameter (490 nm). Differences represent RemoTAP – GRASP. The color bar indicates the number of retrievals.

retrieves a smaller isotropic BRDF parameter than GRASP, it also retrieves a smaller AOD. The difference in AOD increases to -0.10 when the difference in isotropic BRDF is also -0.10. For cases where RemoTAP retrieves a higher isotropic BRDF parameter, there is no clear dependence of AOD differences on difference in isotropic BRDF. There is also a clear dependence of the AOD difference on difference in the BPDF scaling parameter. As notes above, RemoTAP retrieves much smaller values

for the BPDF scaling than GRASP. When the BPDF difference is -8, RemoTAP retrieves on average a higher AOD ( 0.10) than GRASP. These cases correspond mostly to the Sahara and the Arabian Peninsula. On the other hand, when RemoTAP retrieves higher BPDF than GRASP (mostly over higher latitudes), it retrieves smaller AOD, where the mean difference is -0.05. Figure 21 shows the difference in SSA between RemoTAP and GRASP as a function of difference in retrieved isotropic BRDF parameter (at 490 nm) and the difference in BPDF scaling parameter. For SSA, we see in particular a large effect on isotropic

BRDF parameter, where the SSA difference can be almost up to 0.10 when the difference in isotropic BRDF parameter is -0.10. The dependence on BPDF is smaller and mostly apparent when RemoTAP retrieves higher BPDF than GRASP. Figure 20 shows the difference in AE between RemoTAP and GRASP as a function of difference in retrieved isotropic BRDF parameter (at 490 nm) and the difference in BPDF scaling parameter. Here we see a strong dependence on BPDF scaling parameter. For large negative differences, i.e. when RemoTAP retrieves smaller BPDF than GRASP, there is a large positive difference

in AE. These cases correspond mostly to the Sahara where indeed RemoTAP retrieves higher AE than GRASP. Here, the higher polarized reflectance retrieved by GRASP will require bigger aerosol particles (smaller AE) and as a result smaller degree of polarization) over these areas to fit top-of polarized reflectance. In contrast, smaller polarized surface reflectance provided by RemoTAP will result in smaller particles (bigger AE) and larger degree of polarization to insure good fit of




satellite polarization measurements. The AERONET comparison indicates a positive AE bias in RemoTAP over the Sahara,
where a possible explanation is that RemoTAP retrieves a surface reflection matrix with too small polarization. Observed
dependencies require additional studies to improve aerosol size characterization from polarimetric measurements.

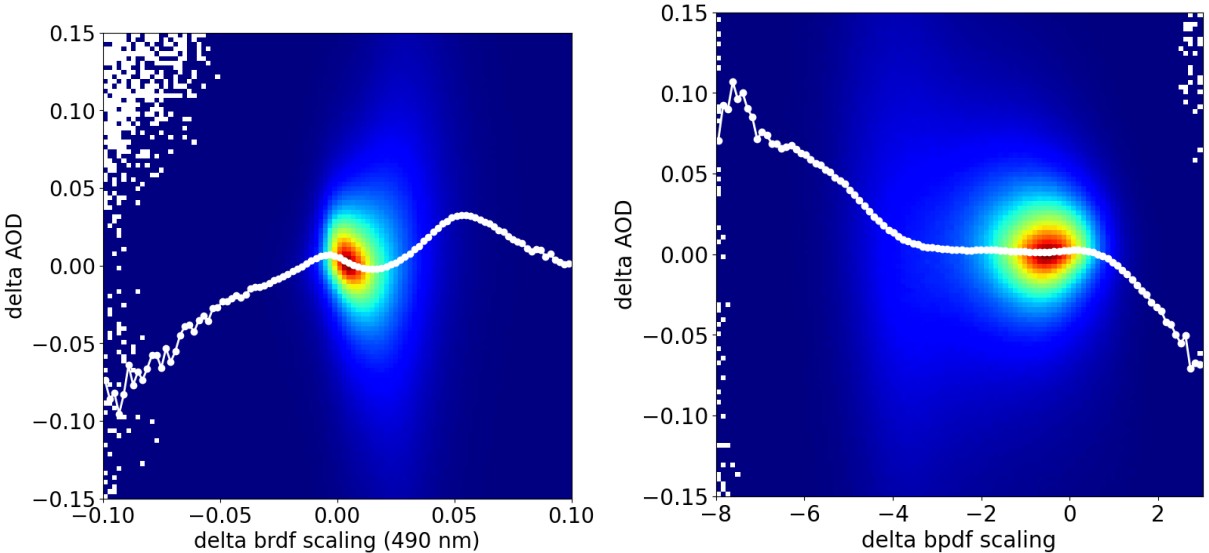

**Figure 19.** Dependence of the AOD difference (RemoTAP – GRASP) on difference in the Isotropic BRDF parameter at 490 nm (left) and
the BPDF scaling parameter (right). The color bar indicates the number of retrievals.

### 4.4.1 Comparison to MODIS

The MODIS Dark Target (DT) (Levy et al., 2013) and Deep Blue (DB) (Hsu et al., 2013) AOD products are the most widely
used aerosol products. Here, we compare AOD and AE retrieved from PARASOL by both RemoTAP and GRASP to the
MODIS DT and DB products. Figure 22 shows the comparison for AOD over land and ocean. Over land, GRASP has smaller
RMSD with MODIS than RemoTAP but it should be noted that the RMSD between GRASP and RemoTAP is smaller (see
Figure 9) than the agreement between both algorithms with MODIS. Over ocean, the agreement of both GRASP and RemoTAP
with MODIS is similar as the agreement between GRASP and RemoTAP, although GRASP has a smaller RMSD with MODIS
than RemoTAP. The smaller differences between GRASP and RemoTAP products over land than the differences between both
algorithms and MODIS may suggest a more accurate AOD product over land for the PARASOL algorithms, especially because
both PARASOL algorithms over land are closer to AERONET than MODIS (Chen et al., 2020). For the global distribution
of differences (not shown), both GRASP and RemoTAP show very similar patterns against MODIS, with a strong positive
difference over most of Africa, India, and China, a weaker positive difference over Europe and the US, and some small spots of
negative differences (e.g. over South America). Over the global ocean, RemoTAP shows a small (0.01-0.02) negative difference





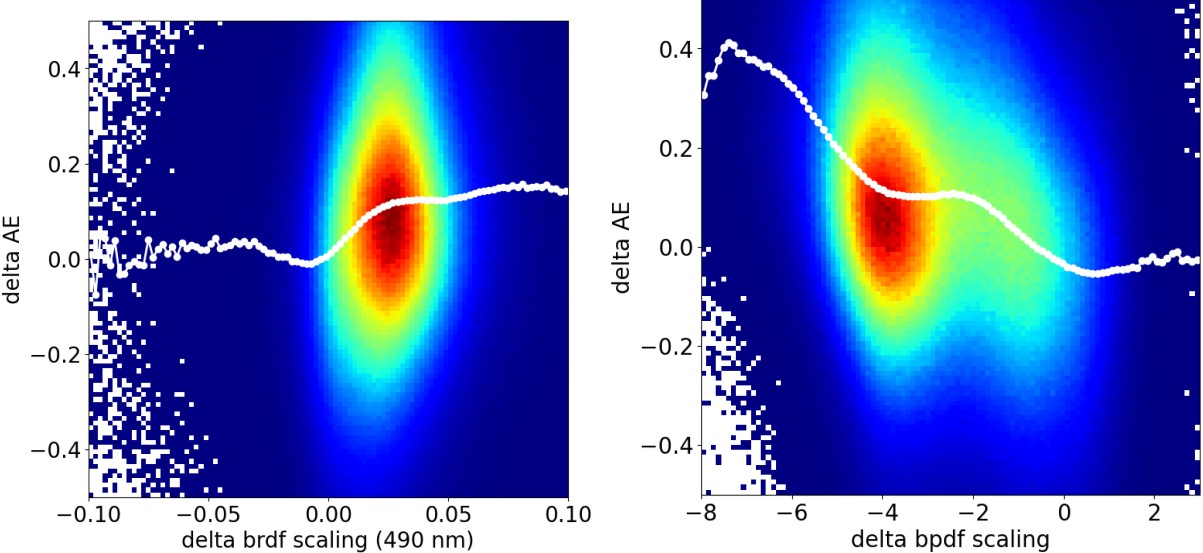

**Figure 20.** Same as Fig. 19 but for AE.

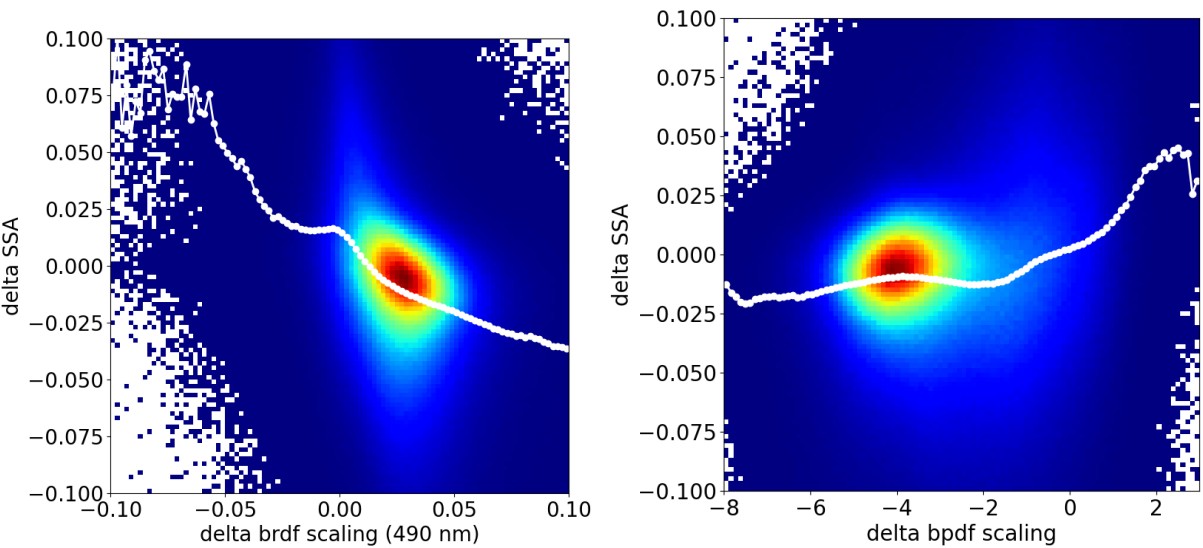

**Figure 21.** Same as Fig. 19 but for SSA.





against MODIS, whereas GRASP shows a small positive difference at mid-latitudes and a small negative difference in most of
the Tropics.

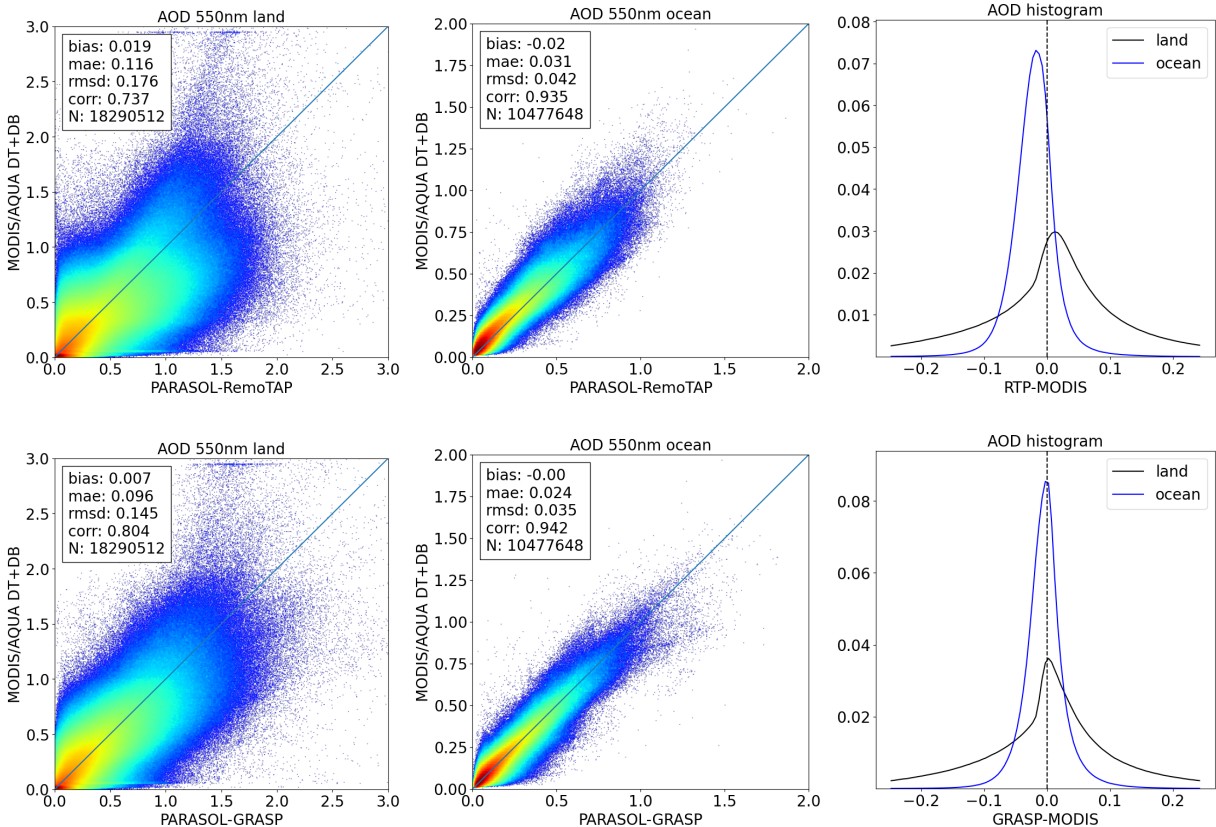

**Figure 22.** AOD Comparison with MODIS for RemoTAP (upper panels) and GRASP (lower panels). Left panels show comparison over land, middle panels comparison over ocean, and right panels histograms of the differences. For MODIS the DT and DB aerosol products have been combined.

Figure 23 shows the AE comparison of both PARASOL products with the MODIS DT product over land and ocean. Over land, we can clearly see that the MODIS-DT AE tends to be centered around a number of discrete values. This is probably a result of the MODIS retrieval approach based a discrete set of aerosol models. Both for RemoTAP and GRASP there is a
large difference with the MODIS-DT AE and the difference between RemoTAP and GRASP, shown in Figure 11, is much smaller than the agreement of both products with MODIS. This is expected because MODIS has limited information content on aerosol size over land which results in poor comparison of MODIS-DT AE against AERONET (Chen et al., 2020). The same conclusions hold for the AE comparison with the MODIS-DB product over land (not shown). Over ocean, the difference between the PARASOL products and MODIS is smaller than over land but still substantially larger than the agreement between



RemoTAP and GRASP. This suggests that also over ocean both PARASOL algorithms provide a more accurate AE than MODIS, as expected.

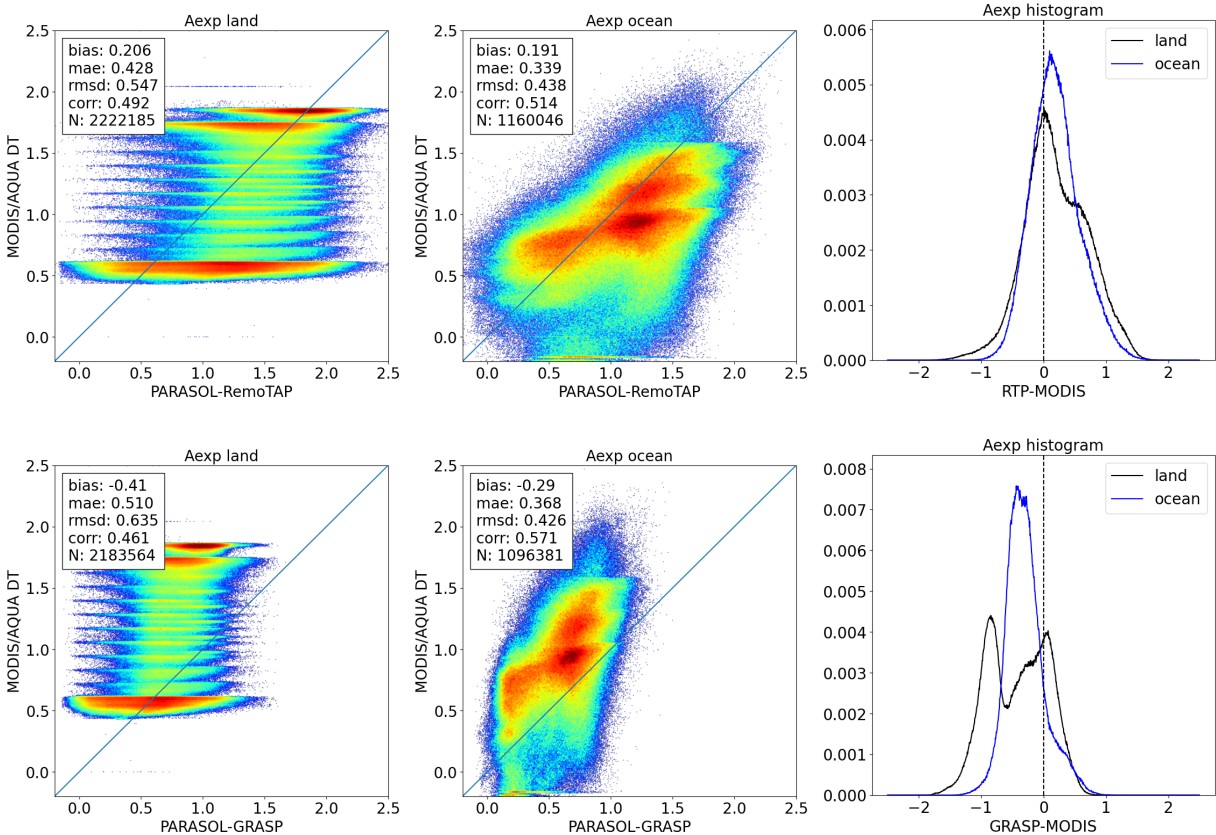

**Figure 23.** AE Comparison with MODIS-DT for RemoTAP (upper panels) and GRASP (lower panels). Left panels show comparison over land, middle panels comparison over ocean, and right panels histograms of the differences.

## 5 Conclusion and Outlook

In this paper, we have performed a systematic intercomparison of the GRASP and RemoTAP algorithms for aerosol retrieval from MAP measurements. The study involved a comparison of forward models, synthetic retrievals, validation results with
AERONET, and a global comparison for 1 year of data processed by both algorithms. The forward models of GRASP and RemoTAP agree within than 2% (1% for most wavelengths and angles) for radiance and within 0.01 (0.005 for most wavelengths and angles) for Degree of Linear Polarization (DoLP). The differences are within the range of PARASOL uncertainties but may become important for more accurate future polarimeters like SPEXone (Hasekamp et al., 2019a) and CO2M (Sierk et al., 2019). Synthetic retrievals were performed on 2 sets of synthetic measurements, one created by the RemoTAP forward model





and the other one by the GRASP forward model. Both sets of synthetic measurements were based on a more complex aerosol description than assumed in the algorithms. Both algorithms perform well on both sets of synthetic measurements, with only better performance for AOD for the algorithm which forward model was used to create the synthetic measurement.

For the AERONET comparisons, we obtained very similar results for AOD retrieved by both RemoTAP and GRASP over land: both algorithms show an RMSE of 0.10 (at 550 nm) and 54% of the retrievals falls within the GCOS requirements. For
SSA, both algorithms show a good performance in terms of RMSE ( 0.04) but RemoTAP has a smaller bias (0.002) compared to GRASP (0.021). For AE, GRASP has a sligtly smaller RMSE (0.367) than RemoTAP (0.387), which is related a small overestimate of AE at low values (large particles) by RemoTAP. Over ocean both algorithms perform very well. For AOD, RemoTAP has an RMSE of 0.057 and GRASP an even smaller RMSE of 0.047. For AE, the RMSE of RemoTAP and GRASP are 0.285 and 0.224, respectively. Based on the AERONET comparison, we conclude that both algorithms show very similar
overall performance, where both algorithms have stronger and weaker points.

Finally, global processing with both algorithms for PARASOL data of the year 2008 has been performed and a comparison for gridded products (0.1 by $0.1^o$) has been performed. For the global comparison of AOD, the RMSD values are 0.12 and 0.037 over land and ocean, respectively. The largest differences occur over the biomass burning region in equatorial Africa. The global mean values are virtually unbiased with respect to each other. For AE the RMSD between RemoTAP and GRASP
is 0.33 over land and 0.23 over ocean when only retrievals with AOD>0.2 are included. When taking retrievals with AOD>0.1, the RMDS increases to 0.48 over land and 0.36 over ocean. Towards lower AOD, significant differences occur between the 2 data products because the AE calculation becomes very sensitive to even small errors in AOD at the different wavelengths when the AOD is small. Concerning fine and coarse AOD, over land, the RMSD is smaller for fine mode AOD (RMSD=0.063) than for coarse mode AOD (RMSD=0.11) and over ocean vice versa (RMSD=0.049 for fine mode and 0.037 for coarse mode). Over
ocean, the fine and coarse mode have opposite biases ( 0.02) that compensate each other in the total AOD. For SSA, we find good agreement over land (RMSD=0.030) for retrievals with AOD > 0.2. Over ocean the agreement is poor with a bias of 0.053 (where RemoTAP retrieves higher SSA) and an RMSD of 0.074. As expected, the differences increase towards low AOD, both over land and ocean. For the AAOD, the agreement is reasonable over lands and biased over ocean, where GRASP retrieves higher AAOD than RemoTAP (as expected from the lower SSA). The surface BRDF products show reasonable agreement,
where for the majority of cases RemoTAP retrieves higher isotropic BRDF parameter than GRASP. The differences in the BPDF are substantial, where RemoTAP retrieves smaller BPDF scaling factor than GRASP in most cases. The differences in AE are strongly related to differences in BPDF.

We also compared the GRASP and RemoTAP AOD and AE products against MODIS. For AOD over land, the agreement of either GRASP or RemoTAP with MODIS is worse than the agreement between the 2 PARASOL algorithms. Over ocean,
the agreement is very similar among the 3 products for AOD. For AE, the difference between GRASP and RemoTAP is much smaller than the difference between MODIS and both products, especially over land. This is expected because the PARASOL measurements have a larger information content on aerosol size than MODIS (e.g. Mishchenko and Travis (1997a); Hasekamp and Landgraf (2007)).





To the best of our knowledge, the agreement between GRASP and RemoTAP is unprecedented. The good agreement between
the 2 products gives confidence in the quality of both products. The agreement of the latest product versions with each other
and with AERONET improved significantly compared to the previous version of the global products of GRASP (Chen et al.,
2020) and RemoTAP (Lacagnina et al., 2017). The results demonstrate that dedicated effort on algorithm development for MAP
aerosol retrievals still leads to substantial improvement of the resulting aerosol products and this is still an ongoing process.
In general, the comprehensive studies performed with GRASP and RemoTAP on synthetic and real PARASOL measurements
allowed identifying weak and strong points of the algorithms, which are crucial for the further improvements of aerosol and
surface characterization from space-borne remote sensing. There are still several aspects that need improvement. Away from
the AERONET stations, differences between RemoTAP and GRASP can be substantially larger, with systematic differences
in a different direction than expected from the AERONET comparison. Most notably, over ocean GRASP retrieves smaller
SSA than RemoTAP. Both GRASP and RemoTAP show at some locations an unexpected sharp transition between land and
ocean for SSA. For Angstrom exponent, both algorithms compare well for moderate and high AOD (>0.2). For low AOD,
retrieval of both AE and SSA becomes more challenging and needs improvement. For AOD, a relatively large bias between
SRON RemoTAP and GRASP occurs over over equatorial Africa. The issue of cloud screening has not been investigated in
the present study. First of all, AERONET measurements have been cloud screened already and hence a comparison between
PARASOL and AERONET involves already an implicit cloud screening. Furthermore, because we have compared RemoTAP
and GRASP for common pixels, effectively the combination of both cloud screening approaches has been used.

Possible points for improvements for MAP aerosol retrieval are improved cloud screening, quality filtering, development of
joint aerosol cloud retrievals (aerosol-above-cloud (Waquet et al., 2014) and in between clouds (Hasekamp, 2010; Stap et al.,
2016)). Also, focused development is needed of higher level data products, such as CCN (Hasekamp et al., 2019b), aerosol
composition (van Diedenhoven et al., 2022; Li et al., 2022b), PM2.5, and radiative fluxes. For validation, a better validation
infrastructure is needed for 'new' aerosol products like Single Scattering Albedo (SSA), size distribution, and refractive index
(chemical composition). AERONET validation of these products are restricted to high AOD cases (> 0.4 at 440 nm) which will
occur less in the coming decade because of expected reduced aerosol emissions. Finally, also for future MAP aerosol retrieval
products, it is important to perform global comparisons between RemoTAP and GRASP, and preferably more algorithms.

*Data availability.* Data are availabe at www.sron.nl/harpol

**Appendix A: Algorithm Descriptions**

**A1  RemoTAP**

The RemoTAP algoritm is based on iterative fitting a forward model to measurements of radiance and/or polarization of
reflected light. The forward model consists of the LINTRAN2 radiative transfer model (Hasekamp and Landgraf, 2005;
Hasekamp and Butz, 2008) and a model to compute optical aerosol properties from micro-physical properties using the Mie-



and T-matrix improved geometrical optics database by Dubovik et al. (2006) along with their proposed spheroid aspect ratio distribution for computing optical properties for a mixture of spheroids and spheres. For retrievals over land RemoTAP uses a surface reflection matrix (Litvinov et al., 2011) that accounts for the directional and polarization properties of the surface, based on the Ross-Li model (Wanner et al., 1995) total reflectance and the model of Maignan et al. (2009) for polarized reflectance. For a full description of the surface reflection model, which is the same for RemoTAP and GRASP, see B1. RemoTAP de-

scribes reflection by the ocean using the a contribution of Fresnel reflection on the rough ocean (Cox and Munk, 1954; Zhang and Wang, 2010) and scattering and absorption inside the ocean body using a neural Network trained with radiative transfer calculations in the ocean body (Fan et al., 2019). Additionally, a wavelength dependent correction term is fitted. For more details see Appendix B2.

While earlier aerosol retrieval studies with RemoTAP used a bi-modal aerosol description (Hasekamp et al., 2011; Wu et al.,

2015; Di Noia et al., 2017) or a 5-mode aerosol description (Fu and Hasekamp, 2018; Fu et al., 2020), in this study we follow the approach of Lu et al. (2022) and describe the aerosol size distribution in RemoTAP by 3 log-normal modes, with one fine mode and 2 coarse modes (soluble and insoluble). The spectrally dependent refractive index $m(\lambda)$ per mode is parameterized by

$$m(\lambda) = \sum_{k=1}^{n_\alpha} \alpha_k \, m^k(\lambda) \tag{A1}$$

where $m^k(\lambda)$ are prescribed functions of wavelength, for which we use standard refractive index spectra for different aerosol components, i.e. Dust (DU) Torres et al. (2001), Inorganic/Sulphate (INORG) and Black Carbon (BC) d'Almeida et al. (1991), and Organic Carbon (OC) (Kirchstetter et al., 2004).

For the fine mode the state vector includes $r_{eff}$, $v_{eff}$, $N_{aer}$ and $f_{sph}$ and the refractive index coefficients $\alpha_k$ that correspond to the standard refractive index spectra 'INORG', 'BC', 'OC'. The coarse insoluble mode consists of non-spherical dust. For

this mode the state vector includes $r_{eff}$, $N_{aer}$, and a coefficient for the imaginary part of the 'DU' refractive index. The fixed parameters are $f_{sph}$, $v_{eff}$, $\alpha_k$ for the 'DU' real part refractive index. One value for $z_{aer}$ is included which is assumed to be the same for modes 1 and 2. The width $w_0$ of the altitude distribution is fixed . The 3rd mode is a coarse soluble mode. For this mode the state vector includes $r_{eff}$, $N_{aer}$, and coefficient $\alpha_k$ of the 'INORG' refractive index spectrum. The fixed parameters are $f_{sph}$, $v_{eff}$, and $z_{aer}$. An overview of the fit parameters is given in Table A1.

## A2   GRASP

The Generalized Retrieval of Aerosol and Surface Properties is a new-generation algorithm developed for deriving extensive aerosol properties from diverse space-borne and ground-based instruments. The bigger the information content is in the remote sensing instrument the higher performance the GRASP algorithm will demonstrate. The overall concept of the algorithm is described by Dubovik et al. (2011) and Dubovik et al. (2021). The algorithm is based on highly advanced statistically optimized

fitting implemented as multi- term least square minimization that had earlier been successfully implemented (Dubovik et al., 2000, 2002; Dubovik et al., 2006) for aerosol retrievals from ground-based AERONET radiometers. The GRASP aerosol model used for global processing in these studies is based on the chemical component approach (Li et al. (2019), (Li et al. (2022a)

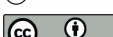



**Table A1.** State vector elements, prior values, and parameter range for the parametric 3-mode aerosol description. The superscript $^{ci}$ and $^{cs}$ denote 'coarse insoluble' and 'coarse soluble', respectively. The 'fit' column indicates whether a parameter is fitted (fit=1) or fixed (fit=0) to the prior value.

| state vector element | prior value | fit | min - max |
|---|---|---|---|
| **Aerosol parameters** | | | |
| $r_{\mathrm{eff}}^{f}$ | 0.15 μm | 1 | 0.02 μm - 0.30 μm |
| $v_{\mathrm{eff}}^{f}$ | 0.20 | 1 | 0.01 - 0.80 |
| $\mathrm{N}^{f}$ | from LUT retrieval | 1 | 0.001 - n/a |
| $f_{\mathrm{sphere}}^{f}$ | 0.95 | 1 | 0.0 - 1.0 |
| $\alpha_{\mathrm{inorg}}^{f}$ | 1.0 | 1 | such that $1.3 < m_r < 1.69$ |
| $\alpha_{\mathrm{bc}}^{f}$ | 0.015 | 1 | such that $1.3 < m_r < 1.69$ |
| $\alpha_{\mathrm{oc}}^{f}$ | 0.10 | 1 | such that $1.3 < m_r < 1.69$ |
| $r_{\mathrm{eff}}^{ci}$ | 1.0 μm | 1 | 0.7 μm - 5.0 μm |
| $v_{\mathrm{eff}}^{ci}$ | 0.60 | 0 | n/a |
| $\mathrm{N}^{ci}$ | from LUT retrieval | 1 | 0.001 - n/a |
| $f_{\mathrm{sphere}}^{ci}$ | 0.0 | 0 | n/a |
| $\alpha_{\mathrm{du,im}}^{ci}$ | 0.95 | 1 | 0-1 |
| $z_{\mathrm{aer}}^{f,ci}$ | 2 km | 1 | -2 km - 10 km |
| $r_{\mathrm{eff}}^{cs}$ | 2.5 μm | 1 | 0.7 μm - 5.0 μm |
| $v_{\mathrm{eff}}^{cs}$ | 0.60 | 0 | n/a |
| $\mathrm{N}^{cs}$ | from LUT retrieval | 1 | 0.001 - n/a |
| $f_{\mathrm{sphere}}^{cs}$ | 1.0 | 0 | n/a |
| $\alpha_{\mathrm{inorg,im}}$ | 0.9 | 1 | such that $1.3 < m_r < 1.69$ |
| $z_{\mathrm{aer}}^{cs}$ | 0.5 km | 0 | n/a |
| **Ross-Li land surface parameters** | | | |
| $A(\lambda_i), i = 1, \cdots, N_{\mathrm{band,map}}$ | from LUT retrieval | 1 | 0.0 - 1.0 |
| $k_{\mathrm{geo}}$ | 0.1 | 1 | 0.0 - 0.35 |
| $k_{\mathrm{vol}}$ | 0.5 | 1 | 0.0 - 1.5 |
| **Land surface parameters (Maignan)** | | | |
| $B$ | 1.0 | 1 | 0.2 - 10.0 |
| $\nu$ | 0.1 | 0 | n/a |
| **Ocean parameters** | | | |
| windspeed | 7 m/s | 1 | 0.1 m/s - 100 m/s |
| $x_{\mathrm{chl}}$ | 2 mg/m$^3$ | 1 | 0 mg/m$^3$ - 30 mg/m$^3$ |
| $A(\lambda_i), i = 1, \cdots, N_{\mathrm{band,map}}$ | 0.0 | 1 | -0.10 - 0.10 |





**Table A2.** State vector description of the GRASP Chemical Components (CC) approach.

| State vector | Initial guess | min - max | Single-pixel constraints (order) | Multi-pixel constraints | |
|---|---|---|---|---|---|
| | | | | Spatial (order) | Temporal (order) |
| Aerosol parameters | | | | | |
| | 0.01 | 0.000005 - 5 | 5.0e-3 (2) | 1.0e-2 (1) | 5.0e-4 (1) |
| | 0.01 | 0.000005 - 5 | –* | 1.0e-2 (1) | 5.0e-4 (1) |
| | 0.01 | 0.0001 - 0.1 | | | |
| | 0.1 | 0.0001 - 0.8 | – | 1.0e-2 (1) | 3.0e-2 (1) |
| | 0.5 | 0.0001 - 0.95 | | | |
| | 0.01 | 0.00001 - 0.03 | | 1.0e-2 (1) | 3.0e-2 (1) |
| | 0.8 | 0.3 - 00.95 | – | | |
| | 0.29 | 0.28 – 0.98 | – | 1.0e-2 (1) | 3.0e-2 (1) |
| | 0.9 | 0.005 - 0.9999 | – | 1.0e-2 (1) | 1.0e-3 (1) |
| | 2000 m | 10 - 5000 m | – | 1.0e-2 (1) | 1.0e-3 (1) |
| Ross-Li land surface parameters | | | | | |
| | 0.1 | 0.001 - [0.2, 0.3, 0.4, 0.7, 0.8, 0.8] | 1.0e-4 (1) | – | 8.0e-0 (1) |
| | 0.1 | 0.01 - 2.0 | 1.0e+1 (1) | – | 8.0e-0 (1) |
| | 0.1 | 0.01 - 1.0 | 1.0e+1 (1) | – | 8.0e-0 (1) |
| Land surface parameters (Maignan) | | | | | |
| | 2.1 | 0.01 - 10 | 1.0e+1 (1) | 1.0e-3 (1) | 8.0e-0 (1) |
| Ocean surface parameters (Cox-Munk) | | | | | |
| | 0.01 | 0.001 - 0.05 | 1.0e-3 (1) | 1.0e-3 (1) | 1.0e-3 (1) |
| | 0.9 | 0.3 - 1 | 1.0e+1 (1) | 1.0e-3 (1) | – |
| | 0.02 | 0.0015 - 0.1 | 1.0e+1 (1) | 1.0e-3 (1) | – |

where aerosol is represented as external mixture of two different aerosol modes. Each aerosol mode itself is considered as internal mixture of different chemical species whose spectral complex refractive index is calculated through Maxwell-Garnett

(MG) effective approximation. The first mode is dominated mainly by small particles whose size is represented by three precalculated log-normal size distributions bins and the complex refractive index is calculated from the mixture of predefined aerosol species: black and brown carbons, quartz as well as ammonium nitrates. The second mode represents coarse aerosol particles with two precalculated log-normal size distributions bins and internbal mixture of iron oxides, quartz, and ammonium nitrates species.

The retrieved parameters of GRASP algorithm are presented in Table A2.



## Appendix B: Land and Ocean Reflection

### B1 Land Surface Reflection

Both RemoTAP and GRASP use the same model to describe the surface reflection matrix over land:

$$\mathbf{R}_s(\lambda, \theta_{\mathrm{in}}, \theta_{\mathrm{out}}, \Delta\varphi) = r_{11}(\lambda, \theta_{\mathrm{in}}, \theta_{\mathrm{out}}, \Delta\varphi)\,\mathbf{D} + \mathbf{R}_{\mathrm{pol}}, \tag{B1}$$

where $\mathbf{D}$ is the null matrix except $\mathbf{D}_{11} = 1$. For the Bi-directional Reflection Distribution Function (BRDF) $r_{11}(\lambda, \vartheta_{\mathrm{in}}, \vartheta_{\mathrm{out}}, \Delta\varphi)$ the Ross-Li model modified for Hot Spot effect is used Maignan et al. (2009). Moreover, following the studies performed in Litvinov et al. (2011), $r_{11}$ is renormalized on the isotropic parameter to separate spectral and angular dependent terms:

$$r_{11}(\lambda, \theta_{\mathrm{in}}, \theta_{\mathrm{out}}, \Delta\varphi) = A(\lambda)\,(1 + k_{\mathrm{geo}} f_{\mathrm{geo}}(\theta_{\mathrm{in}}, \theta_{\mathrm{out}}, \Delta\varphi) + k_{\mathrm{vol}} f_{\mathrm{vol}}(\theta_{\mathrm{in}}, \theta_{\mathrm{out}}, \Delta\varphi)) \tag{B2}$$

where $f_{\mathrm{geo}}$ and $f_{\mathrm{vol}}$ are respectively the geometric (Li-Sparse) and volumetric (Ross-Thick, modified for Hot Spot effect)
kernels (Wanner et al., 1995, and references therein). They are given by

$$f_{\mathrm{vol}}(\theta_{\mathrm{in}}, \theta_{\mathrm{out}}, \Delta\varphi) = \frac{(\pi/2 - \gamma)\cos\gamma + \sin\gamma}{\mu_{\mathrm{in}} + \mu_{\mathrm{out}}} f_{HotSp} - \frac{\pi}{4} \tag{B3}$$

$$f_{\mathrm{geo}}(\theta_{\mathrm{in}}, \theta_{\mathrm{out}}, \Delta\varphi) = O(\theta'_{\mathrm{in}}, \theta'_{\mathrm{out}}, \Delta\varphi) - \sec\theta'_{\mathrm{out}} - \sec\theta'_{\mathrm{in}} + \frac{1}{2}(1 - \cos\Theta')\,\sec\theta'_{\mathrm{out}}\,\sec\theta'_{\mathrm{in}} \tag{B4}$$

$$O = \frac{1}{\pi}(t - \sin t \cos t)(\sec\theta'_{\mathrm{out}} + \theta'_{\mathrm{in}}) \tag{B5}$$

$$\cos t = \frac{h}{b}\frac{\sqrt{D^2 + (\tan\theta'_{\mathrm{in}}\,\tan\theta'_{\mathrm{out}}\,\sin\Delta\phi)^2}}{\sec\theta'_{\mathrm{out}} + \sec\theta'_{\mathrm{in}}} \tag{B6}$$

$$D = \sqrt{\tan^2\theta'_{\mathrm{in}} + \tan^2\theta'_{\mathrm{out}} - 2\,\tan\theta'_{\mathrm{in}}\,\tan\theta'_{\mathrm{out}}\,\cos\Delta\phi} \tag{B7}$$

$$\cos\Theta' = -\cos\theta'_{\mathrm{in}}\,\cos\theta'_{\mathrm{out}} - \sin\theta'_{\mathrm{in}}\,\sin\theta'_{\mathrm{out}}\cos\Delta\phi \tag{B8}$$

$$\theta'_{\mathrm{out}} = \tan^{-1}\left(\frac{b}{r}\,\tan|\theta'_{\mathrm{out}}|\right), \tag{B9}$$

$$\theta'_{\mathrm{in}} = \tan^{-1}\left(\frac{b}{r}\,\tan|\theta'_{\mathrm{in}}|\right), \tag{B10}$$

$$f_{HotSp} = 1 + \frac{1}{(1 + \pi - \gamma)/\alpha + 0}, \qquad \alpha_0 = 1.5. \tag{B11}$$

Numerically, the absolute value of the RHS of Eq. B6 can exceed 1. In this case we set $\cos t = 1$. The volumetric kernel represents the scattering within a dense vegetation canopy, and is based on a radiative transfer approximation of single scattering due to small, uniformly distributed and non-absorbing leaves. The angular behavior of this kernel is to have a minimum near the backscatter direction and bright limbs (Knobelspiesse et al., 2008). The geometric kernel represents surfaces with larger gaps between objects, and thus accounts for self shadowing. The angular behavior of this kernel is therefore to have a maximum





at backscattering where there are no shadows. $f_{\text{geo}}$ is based on the work of Wanner et al. (1995); Li and Strahler (1992), but is used in the reciprocal form given in Lucht et al. (2000), for the case that the ratio of the height of the tree at the center of the crown to the vertical crown radius, $h/b = 2$, and the ratio of the vertical crown radius to the horizontal crown radius is (spherical, or compact crowns) $b/r = 1$.

In Eq. (B1) $\mathbf{R}_{\text{pol}}$ is given by Maignan et al. (2009)

$$\mathbf{R}_{\text{pol}}(\theta_{\text{in}}, \theta_{\text{out}}, \phi_v - \phi_0) = B_{\text{pol}} \left( \frac{\exp\left(-\tan(\frac{\pi - \Theta}{2})\right) \exp\left(-\nu\right) \mathbf{F}_p(m, \Theta)}{4(\mu_{\text{in}} + \mu_{\text{out}})} \right). \tag{B12}$$

Here, $B_{\text{pol}}$ is a scaling parameter (band-independent). $\mathbf{F}_p(m, \Theta)$ is the the Fresnel scattering matrix with refractive index $m = 1.5$. We use $\nu = 0.1$ (Litvinov et al., 2011).

## B2    RemoTAP Ocean Reflection

For retrievals over ocean, RemoTAP describes the ocean reflection matrix as

$$\mathbf{R}_s(\lambda, \theta_{\text{in}}, \theta_{\text{out}}, \Delta\varphi) = \mathbf{R}_{frn}(\theta_{\text{in}}, \theta_{\text{out}}, \Delta\varphi) + \mathbf{R}_{ul}(\lambda, \theta_{\text{in}}, \theta_{\text{out}}, \Delta\varphi) + A(\lambda) \mathbf{D} \tag{B13}$$

where $\mathbf{R}_{frn}$ is the contribution of the ocean surface, which is described by Fresnel reflection on a rough ocean surface, depending on the wind speed- and direction to provide a Gaussian distribution of surface slopes (Cox and Munk, 1954). $\mathbf{R}_{ul}$ is the ocean body (underlight) contribution. For the ocean body, we need a bio-optical model to compute optical properties of the ocean from bio-physical ocean parameters. We used the bio-optical model of Chowdhary et al. (2012) for case-1 waters

(open ocean) that has the chlorophyll-a concentration $x_{\text{chl}}$ as the only bio-physical ocean parameter to compute the ocean optical properties (single scattering albedo, phase matrix). Using the hydrosol model of Chowdhary et al. (2012), the ocean is described as a mixture of sea-water and a particulate component. The scattering and absorption coefficients of sea-water are taken from Smith and Baker Smith and Baker (1981), while the optical properties of the particulate components were calculated using detritus–plankton (D–P) mixtures. The particulates were assumed to be spherical, so the scattering phase matrix could be

obtained using Mie calculations. The relative contribution of detritus and plankton are parameterized by $X_{\text{chl}}$. Here it should be noted that the underlight contribution is insensitive to the optical depth of ocean when the ocean optical thickness is larger than 10. In this study, we set the ocean optical depth to 20 and assumed a black ocean bottom surface. This ocean surface/body system was being solved by a vector radiative transfer model Hasekamp and Landgraf (2002); Schepers et al. (2014). However, this model is computationally expensive because of the large ocean optical thickness. As an alternative, a Neural Network

(NN) has been designed to simulate (Fourier coefficients of) the ocean body contribution to the reflection matrix just above the ocean surface, with as input the oceanic chlorophyll-a concentration. Finally, $A(\lambda)$ in Eq. (B13) is a wavelength dependent Lambertian albedo term that accounts for oceanic foam but may also correct for errors in $\mathbf{R}_{ul}$. $W_s$, $X_{\text{chl}}$, and $A(\lambda)$ for the different spectral bands are included in the state vector.





## B3   GRASP Ocean Reflection

The reflection matrix over water surfaces ($\mathbf{R_s}$) in GRASP algorithm, used in these studies, is represented as follows:

$$\mathbf{R}_s(\lambda, \theta_{\text{in}}, \theta_{\text{out}}, \Delta\varphi) = \mathbf{R}_{frn}(\theta_{\text{in}}, \theta_{\text{out}}, \Delta\varphi)\delta_{frn}f_{shad} + r_0(\lambda)\mathbf{D} \tag{B14}$$

where $\mathbf{R}_{frn}$ is Fresnel reflection matrix from water surface facets with refractive index $m$, $f_{shad}$ is the shadowing function for Gaussian random rough surface (Tsang et al. (1985); Mishchenko and Travis (1997b)), $\sigma^2$ is the mean square facet slope related to wind speed Cox and Munk (1954), $\delta_{frn}$ is fraction of water surface providing Fresnel reflection, $r_0(\lambda)$ is the isotropic

spectrally dependent water leaving reflectance. In the presented studies no a priori information about wind speed was used and three parameters $\sigma^2$, $\delta_{frn}$ and $r_0(\lambda)$ were retrieved with application of proper spatial and temporal constraints Table A2.

*Author contributions.* OH, PL and OD designed the research. GF, CC, OH, PL performed the research. OH write the first papers draft and all authors contributed to the final version through comments and writing of sub-sections.

*Competing interests.* At least one of the (co-)authors is a member of the editorial board of Atmospheric Measurement Techniques


*Acknowledgements.* The research of this paper has been performed in the framework of the project HARPOL (Harmonizing and advancing retrieval approaches for present and future polarimetric space-borne atmospheric missions), funded through the ESA program EO Science for Society. We acknowledge Christian Retscher (ESA) for his work as Technical officer for HARPOL. We also acknowledge Daniele Gasbarra (ESA) and Alexandru Dandocsi (ESA) for important feedback during the course of the HARPOL project.



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
