# Peer review of "Algorithm evaluation for Polarimetric Remote Sensing of Atmospheric Aerosols"

_Atmospheric Measurement Techniques, 2023_

## Author Response (AR1)

**Reviewer 1**

*The authors evaluate the performance of the most recent version of two global aerosol algorithms, GRASP and RemoTAP, applied to POLDER/PARASOL onboard the A-Train, a space-borne sensor that measured intensity and polarization of backscattered light at multiple wavelengths and viewing angles.*

***Major comments***

*This paper is very well structured and written. The scientific analysis is robust, the results comprehensive are clearly laid out. Such a study and its results are essential to pave the way for the next generation of spaceborne aerosol and cloud missions that will include a polarized sensor (e.g., HARP/ SPEXOne on PACE, 3MI, AOS etc.).*

We would like to thank the reviewer for the important comments and corrections.

*We recommend that the authors:*

1. *Clearly spell out, define, and consistently use their abbreviations throughout the text for all recent vs. older versions of GRASP and RemoTAP (e.g., GRASP-CC vs. GRASP-O, -HP and -M)*

We have corrected this in the revised version

2. *Discuss how they were able to validate their spaceborne aerosol properties using AERONET stations over ocean. They must have used coastal AERONET stations, in which case they might want to discuss the performance of their spaceborne observations over complex surfaces and how this would not be an issue for MAP compared to spaceborne sensors such as MODIS/ VIIRS*

We added explicitly that we validate ocean retrievals with coastal stations and also add the phrase "It should be noted that in coastal regions the ocean (body)reflection is more complex than over the open ocean, which may be expected to result in less accurate aerosol retrievals".

We are not sure if this is less of an issue for MAP retrievals compared to MODIS/VIIRS.

3. *Should briefly discuss their choice of aerosol type in Table 1 i.e., with an RRI of 1.54 and an IRI of 0.005 more representative of maybe mineral dust. Same*

*comment applies to Table 3 – why choose these AERONET stations? Maybe they provide a wide enough range of geometries*

This refractive index is mostly representative for organic carbon. We added some extra discussion on the choice of aerosol model:

"So, different size distributions are accounted for by different contributions of the fine- and coarse mode. We do not expect the forward model comparison to depend much on refractive index because the  dependence of optical properties on the refractive index (in the Mie/T-matrix code) is treated the same in GRASP and RemoTAP."

4. *It is not clear why the authors compare POLDER and MODIS products (section 4.4.1) under section 4.4 (i.e., surface properties)*

This was by accident (thanks for spotting) and is corrected in the revised version.

**Detailed comments:**

*Abstract: The authors need to spell out MAP once*

Corrected in revised version.

*Line 57: Maybe add "from space" after "the only MAP instrument…"*

Added 'space-based'

*Line 108: SPEX and airMSPI need to be spelled out.*

Corrected in revised version

*Line 151: "includes" is repeated twice*

Corrected in revised version

*Line 154: "a posterior filter removing retrieval"*

Changed to "a posterior filter removing pixels for which"

*Line 163-169: See major comment #1 above*

We now mention explicitly the different GRASP versions (CC, HP, M) throughout the manuscript.

*Line 173: We suggest "we use the set of aerosol properties provided in Table 1 (…) in Table 2". Also see major comment #3 above.*

We have modified it according to the response to major comment #3.

*Line 200: based on direct sun*

Corrected

*Line 203: Doesn't the AERONET SSA accuracy depend on the AOD range?*

The 0.03 uncertainty is given by Dubovik et al. (2000). This reference is added to the paper. This is for the level-2 product that indeed only contains cases where AOD (440 nm) > 0.4.

*Line 230: The reader could use a reminder of the POLDER-3/PARASOL measurement uncertainty*

Added "(~2\% for radiance, 0.01-0.02 for DoLP)"

*Line 244: This is the first time in the text that ECHAM is called "SRON-ECHAM".*

Replaced by "RemoTAP-ECHAM"

*Line 263: Tables are often introduced in the order that they are numbered (e.g., Table 4 before 5)*

Moved the table with results from previous algorithm versions ahead.

*Line 266 and legend of Table 4: The reader could use a reminder of the current spatial gridding in this study (0.1º)*

Reminder added

*Line 268-69: See major comment #1 above*

We now mention explicitly the different GRASP versions (CC, HP, M) throughout the manuscript.

*Section 4.1.2: See major comment # 2*

See our response to major comment #2

*Figure 7: Add "over ocean"*

Done.

*Figure 8: The reader might wonder which one is right/ wrong when looking at the AOD, AE or SSA bias between RemoTAP and GRASP. Would it be possible to quickly refer to the*

*AERONET evaluation results for a few stations as an illustration (like it is done for Fig. 10)?*

The problem is that the largest bias occurs over ocean where we have only a very limited number of validation points for SSA.

*Figure 9, 11, 13, 22: blue and black are not easy colors to differentiate in the histogram plot.*

Modified to solid and dashed lines

*Figure 12 and 17: y-axis says "sron" instead of RemoTAP*

Corrected

*Line 347: "that" is repeated twice*

Corrected

*Line 385: "significantly"*

Corrected

*Line 410: "as noted"*

Corrected

*Line 427: authors might want to discuss potential "ground truth" for such spaceborne BRDF/ BPDF retrievals*

We added "Having a "ground truth" for BRDF and BPDF retrievals, e.g. from airborne campaigns (Litvinov et al., 2011) would be useful for this purpose."

*Figure 20, 21: "Same as Fig. 19"*

Corrected

*Line 429: A few words explaining why it is important here to compare POLDER-derived to MODIS-derived aerosol properties would help.*

We believe that the statements that "The MODIS Dark Target (DT) (Levy et al., 2013) and Deep Blue (DB) (Hsu et al., 2013) AOD products are the most widely used aerosol products" already provides an explanation for this.

*Line 431: authors might want to say how they combined DT and DB from MODIS*

We use the MODIS C6.1 megared or combined DT and DB dataset that is with the "AOD_550_Dark_Target_Deep_Blue_Combined" SDS name in the Level 2 product. By taking advantage of DT and DB algorithms, the combined product can provide the best spatial coverage which is important for intercomparison. The details of the merge procedure and overall performance of this merged dataset can be found in Sayer et al. (2014).

*Line 445: "based on a discrete"*

I could not spot the typo...

*Line 448: you might want to also refer to Reid et al., 2022; Reid, Jeffrey S., et al. "A coupled evaluation of operational MODIS and model aerosol products for maritime environments using sun photometry: evaluation of the fine and coarse mode." Remote Sensing 14.13 (2022): 2978.*

Added reference to Reid et al.

*Line 459: spell out CO2M*

Done

Line 462-3: sentence is a little too convoluted

*Removed last part of the sentence.*

*Line 467: "related to a small"*

Corrected

*Line 514: authors might want to add aerosol retrievals below thin clouds as a future research product*

Done.

*Line 518: authors might want to ask for more aerosol validation over ocean, where we do not have AERONET stations (e.g., airborne lidar and/ or sunphotometer)*

Added a sentence "Also, more validation points for retrievals over ocean are needed."

**Reviewer 2**

*In this manuscript titled "Algorithm evaluation for Polarimetric Remote Sensing of Atmospheric Aerosols" by Hasekamp et al, two advanced retrieval algorithms which exploit the rich aerosol information from measurements with multiple angles, wavelengths and polarizations are compared based on both synthetic data and global scale PARASOL data. Collocated AERONET data are used to evaluate the retrieval qualities. These two algorithms are the Generalized Retrieval of Atmosphere and Surface Properties (GRASP) algorithm and the Remote Sensing of Trace gas and Aerosol Products (RemoTAP) algorithm. Both are well received in the community and widely used with broad applications.*

*Overall, good agreements are found between RemoTAP and GRASP, and between the individual algorithms with AERONET data product. The differences in the RemoTAP and GRASP algorithms, at different regions, AOD ranges, etc, are also discussed. The comparison between the two algorithm is important for the community to understand the advantage of the polarimetric remote sensing and impacts from the assumptions in the aerosol and surface properties. Therefore, this work provides useful guidance for the future improvements. The manuscript well written with comprehensive information on the algorithm, synthetic and real observations, comparison details of key aerosol optical properties, such as AOD, SSA, AE etc. I would recommend the publication of this manuscript after addressing a few suggestive comments for clarities.*

We would like to thank the reviewer for the important comments and corrections.

*General comments:*

1. *It is nice to see the converge of the two leading retrieval algorithms in many of the aerosol properties. However, it is not clear whether the retrieval uncertainties and differences are due to inversion algorithm (cost function, convergence, numerical accuracy of forward model etc), or measurement and forward model uncertainty assumptions. Can the author clarify whether the ideal retrieval uncertainties from error propagation have been derived and compared? Such comparisons can also help us to understand the sources to explain the differences between the two algorithms, and any potential to improve on the properties where both algorithms already agree on.*

For the products derived in this study, formal retrieval uncertainties are not available, but they will be in future versions of the products. We added in the conclusion of the revised manuscript: "For future algorithm comparisons it will be

useful to take into account formal retrieval uncertainties based on error propagation."

> 2. *It seems different cost functions are used in the inversion. RemoTAP relies on the reflectance and DoLP, and GRASP replies on reflectance, q, and u (Page 5, Line 149). Different uncertainties are also used in DoLP and q,u. Since the cost function is fundamental in the inversions, can the authors comment that how much impacts of the choice of these different cost functions?*

The impact of using DoLP instead of q and u is not large (we have tested both options with RemoTAP in the past). The impact of the assumed uncertainties can potentially be large for the individual algorithms but the impact is algorithm-dependent. Also, the impact of the assumed uncertainty can be partly compensated by the a posteriori filtering approach. In this work we have chosen to optimize both algorithms individually (combination of assumed uncertainty and filtering). We added in the revised manuscript the following:

"The different filtering approaches between GRASP and RemoTAP can partly compensate for the different assumed measurement uncertainties."

> 3. *It seems GRASP algorithms use a multi-pixel approach with additional constraints on spatial and temporals, can the authors clarify where are the comparisons between single pixels retrievals and where are with multi-pixel retrievals? Can you quantify how much the multi-pixel algorithm helps the aerosol retrievals and how that impacts this work?*

We added the sentence "All GRASP retrievals in this paper have been performed using the multi-pixel approach."

We did not investigate the difference between multi-pixel and single-pixel GRASP retrievals.

*Detailed comments:*

*1.Page 3, Line 80: "Of the full inversion approaches only the RemoTAP and GRASP algorithms have demonstrated capability at a global scale. "*

*Among the algorithm mentioned in the manuscript, it seems MAPOL has also been used in a global scale processing but on synthetic data (https://egusphere.copernicus.org/preprints/2023/egusphere-2023-1843/).*

We mean here actually global processing of real data so made that explicit now. Also added the reference to the latest MAPOL paper.

*2.Is processing resource a factor which impact the capability to process global scale data? can you provide an estimate of the resources used between these two algorithms to process the whole year of PARASOL data?*

Both GRASP and RemoTAP have similar computational cost. We did not compare exact numbers because different facilities have been used for processing. The cost is between 1-2 seconds per pixel per thread.

*3.Page 3, Line 88: "although based on theoretical information content a better agreement is expected".*

*Similar to the general comments, can you provide more information on the gaps and possible causes?*

Based on your comment, we realize that this statement is inaccurate as we were not able to directly link the performance against AERONET (of the latest versions of this paper as well as for previous versions) to the expected theoretical performance for the same data set. Therefore, we modified the statement to "...although also limitations of the products were identified in the validations studies (e.g. bias at low AOD for both products)."

*4. Page 4, line 118: "This allows to improve retrieval by using additional a priori constraints on spatial or temporal variability of any retrieved parameter in different pixels", also later stated in Page 5, Line 127.*

*Also related to the general comments, can you confirm whether such multi-pixel approach is used in this study, and how that impact this study?*

See our response to the general comment.

*5. Page 5, Line 152, " The assumed uncertainty is 1% on I and 0.007 (absolute) on DoLP "
(for RemoTAP), and "The assumed uncertainty is 1 % on I and 0.002 (absolute) on q and
u. " (for GRASP).*

*Why is there higher accuracy for q and u, and how that impacts retrieval results?*

These are different assumptions used in RemoTAP and GRASP, i.e. GRASP assumes the polarization measurements are more accurate than RemoTAP. As noted above, the effect of this assumption can (partly) be compensated by a different filtering approach (see also our response to the general comment).

*6. Page 6, Line 162: "The version of GRASP is based on the recently developed Chemical Component (CC) approach (Li et al., 2019; Zhang et al., 2021), while the previous global versions of GRASP (Chen et al., 2020) were based on the GRASP "Optimized", "High Precision", and "Models" approaches, referred to as GRASP-O, GRASP-HP, and GRASP-M."*

*The authors later mentioned different performance of these algorithms. Does it due to a better choice of priori constraints in the aerosol models? Better alignment with assumptions in AERONET inversions?*

The GRASP-CC approach used in this paper shows the best comparison to AERONET. Most likely this is explained by the better a priori constraints (making use of known spectral dependence of component refractive indices). The GRASP-CC approach is not better aligned with AERONET assumptions than the other approaches.

*7. Page 8, line 201: "Here, it should be noted that the AOD (and hence AE) AERONET product is based om direct sun measurements achieving high AOD accuracy (0.01-0.02) whereas the SSA is based on inversion of diffuse sky measurements which rely on several retrieval assumptions leading to a moderate accuracy of (0.03)."*

*Thanks for mentioning the AERONET product uncertainty. It seems both GRASP and RemoTAP demonstrated a similar RMSE in the order of 0.03-0.04 (Table 6 and Table 8) as comparing with the AERONET data. Since the value is so close to the AERONT uncertainty, can the difference come from the AERONET data itself? It would be very interesting to understand whether such accuracy is at the limit of the information content of the PARASOL data itself.*

*Similarly, to the AOD as well, as both algorithms seems have a larger RMSE in the order of 0.1, while AEROENT accuracy is 0.01-0.02. It seems a lot room to improve the retrieval product if there is enough information.*

In the conclusion we added:

"It should be noted that for SSA the agreement with AERONET in terms of RMSE is already close the the SSA uncertainty of AERONET itself (0.03), so possible further improvements of the algorithms for SSA will be hard to evaluate with AERONET. On the other hand, the RMSE for AOD is much larger than the AERONET uncertainty (0.01-0.02), which means there is enough margin to evaluate further improvements."

*8. Page 8, line 223, "Figure 1 shows the forward model comparisons between RemoTAP and GRASP. The bias and standard deviation of the differences in radiance are mostly below 2% and for DoLP mostly below 0.005."*

*Can the authors comment about whether the differences are coming from physical models or also related to numerical accuracy in the RT simulations?/ geometric optics calculations*

We suspect the main difference the forward models comes from differences in the way optical aerosol properties are computed from micro-physical properties using the tabulated Mie / T-matrix calculations. See the end section 3.1. The RT solvers have both been extensively tested against independent references do not introduce significant errors (see also our remark in section 2.1)
*Since GRASP use q and u in the retrievals, are their uncertainties also quantified?*

GRASP assumes an uncertainty of 0.002 on both q and u as stated in section 2.1

*9. Page 9, Figure 1: the plot seems noisy, do you add noise to the simulation?*

We did not add noise to the simulations. The noisy appearance comes from inconsistency between assumptions made in the aerosol retrieval and in the synthetic measurement creation, where the latter uses a more detailed and realistic aerosol description.

*10. Page 10, Figure 2: "Results have been filtered for RemoTAP chi2 < 1 and GRASP minimum residual< 3% ", why GRASP only report a residual?*

Both GRASP and RemoTAP made different choices in their filtering approach. The combination of assumed uncertainty and filtering approach results in a given performance. The GRASP team found that for the GRASP algorithm a filter based on fit residual leads to the best balance between accuracy and data yield.

*Why RemoTAP choose chi2<1? For chi2<1, it would suggest overfitting, or the uncertainties used in the retrieval is underestimated.*

Given that these are simulations without noise, in this specific situation chi2<1 does not mean overfitting.

*Does GRASP also use a chi2 type of cost function? Choosing a minimum residual < 3%, would lead to a chi2 cost function value of (3%/1%)^2 = 9 for reflectance? Is this a correct estimation?*

GRASP uses relative Root Mean Square Error to estimate difference between measurements and fit (measured reflectance and fitted reflectance). Minimum residual < 3% means that this relative RMSE <3%

*11. Page 14, Fig 4, Page 17, Fig 7:*

*It seems there is not much sensitivity for AOD<0.1. What is the possible reason? I would expect a higher accuracy due to the high accuracy in the reflectance measurements (1%).*

The log-scale of course emphasizes differences at small AOD. We did not evaluate the range 0-0.1 for AOD but the metrics for the range 0-0.2 indicate reasonable performance, with 76.6% (RemoTAP) and 83.8% within the GCOS requirement.

*12. Page 17, Table 7:*

*It is nice to report the uncertainty with respect to the AOD ranges! Is there such table for retrievals over land?*

Yes, please see Table 6.

13. Page 32, Line 521: it seems only RemoTAP data is available and GRASP data access need permissions.

The access permission is now granted for everyone.

*14. Page 35, Table A2: There are a long list of parameters, but it is not mentioned what they are?*

Corrected

**Reviewer 3**

**Review of "Algorithm evaluation for Polarimetric Remote Sensing of Atmospheric Aerosols"**

**General comments:**

*This study evaluates the performance of the two popular multi-angle polarimetric aerosol retrieval algorithms GRASP and RemoTAP using both synthetic data and PARASOL real measurements. Results show that retrievals from the two algorithms are not only in agreement with each other but also validated well against AERONET data. This manuscript is well organized.*

 We would like to thank the reviewer for the important comments and corrections.

**Specific comments:**

1.  *The largest AOD difference between GRASP and RemoTAP exist over Sahara/Arabia and equatorial Africa. I suggest discussing the possible reasons. Are surface reflectance differences also large at these regions.*

This is discussed in relation to Fig. 19: "When the BPDF difference is -8, RemoTAP retrieves on average a higher AOD ( 0.10) than GRASP. These cases correspond mostly to the Sahara and the Arabian Peninsula."  In section 4.2.1 we now added the phrase "…related to differences in retrieved BPDF (see section 4.4)"

For equatorial Africa we do not have a good explanation.

2.  *The caption of Fig. 4 should be put below Fig. 4.*

This is corrected in the revised version.

3.  *Colorbar are missing in many figures (Figs. 9, 11, 12, 13, 14, and 16 - 23).*

We added color bars to the 'gridded' figures (17-21) that show dependencies. For the scatter-plots we did not add color bars because here the role of the colors is to indicate the data density and we believe a color bar does not provide useful information.

**Reviewer 4**

**General Comments:**

*This manuscript presents a thorough evaluation of two prominent multi-angle polarimetric aerosol retrieval algorithms, GRASP and RemoTAP, utilizing both synthetic data and real measurements from the PARASOL satellite. The study demonstrates that the results obtained from these two algorithms not only exhibit strong agreement with each other but also show robust validation against AERONET data. The manuscript is well-structured and effectively communicates the scientific analysis and comprehensive results. Such research is of great significance, laying the foundation for forthcoming spaceborne aerosol and cloud missions equipped with polarized sensors. Therefore, it is recommended for the acceptance of this study after the following issues have been addressed.*

We would like to thank the reviewer for the important comments and corrections.

**Specific Comments:**

*Abstract: The full names of several abbreviations should be introduced, such as of "MAP". Similar issues are also present in the main text (SRON, CO2M, etc.).*

We included the meaning of MAP and CO2M. 'SRON' is not an abbreviation, but SRON Netherlands Institute for Space Research is the full name.

*Line 151: There are two "includes".*

Corrected

*Figure 1: Using different line styles in the figures is preferable to using different colors (e.g., in Figure 5).*

We believe that in case of Fig. 1, colors work well because there are just 2 lines, that can be easily distinguished in this figure.

*Table: Tables should be formatted as three-line tables whenever possible.*

Personally, I prefer more separators, but I guess this needs to be resolved in the type-setting stage according to AMT standards.

*Figure 4: The figure caption should be placed below the figure.*

Corrected in revised version.

*Figure 9, 11, 13, 14, 16, 22: Same issues with in Figure 1 mentioned above.*

We changed the figure such that land and ocean are indicated with solid and dashed lines, respectively.

*Line 346: There are two "that".*

Corrected

**Reviewer 5**

*This important paper concerns the important topic of polarimetric remote sensing, which represents the new state-of-the-art in passive remote sensing of the Earth's atmosphere-ocean/surface system. This paper compares two advanced and powerful polarimetric aerosol remote sensing retrieval algorithms, namely GRASP and RemoTAP, as applied to simulated polarimeter data, PARASOL/POLDER-3 satellite polarimeter data, and with comparisons made against AERONET ground-based aerosol retrievals and MODIS aerosol products (non-polarimetric aerosol retrievals). The paper is well-written and logically structured. The findings in this paper advance the state-of-the-art in polarimetric aerosol remote sensing, and the paper is complete in terms of content.*

 We would like to thank the reviewer for the important comments and corrections.

*The paper is thus recommended for publication subject to minor revisions to address the following recommendations and comments:*

*Abstract: The term "good agreement" is qualitative. For example, an RMSD difference of 0.12 in AOD at 550 nm (or 0.1) does not seem "good". What does "good agreement" mean? Does it mean to achieve as much agreement as can be expected between RemoTAP and GRASP aerosol and surface properties retrieved for a given instrument (e.g. PARASOL/POLDER-3) with its particular set of channels, angles and corresponding instrument measurement uncertainties?*

We agree and removed these qualitative statements from the abstract.

*Line 150: Justify/cite the choice of 1% radiometric measurement uncertainty. Is this a 1-sigma (one standard deviation) uncertainty?*

We added the following in the revised version:

"The assumed measurement uncertainties for both GRASP and RemoTAP are likely underestimated (Fougnie et al. 2007,Snik et al, 2014)  which results in chi2 values that are larger than one."

and

"The different filtering approaches between GRASP and RemoTAP can partly compensate for the different assumed measurement uncertainties."

These are indeed 1-sigma uncertainties as indicated in the revised version.

*Line 230: Add a reference for the PARASOL/POLDER-3 instrument uncertainties that are mentioned.*

References have been added in the revised manuscript.

*Section 4.4 (Surface properties). Distinguish between surface properties over land vs ocean. Is there evidence to suggest that the differences between the ocean surface models in RemoTAP and GRASP can be responsible for the considerable differences in retrievals of aerosol SSA over the ocean? Or is it something else?*

We only compared surface properties over land. To make that clear we rename the corresponding subsection in the revised manuscript as "Land Surface Properties". GRASP and RemoTAP use different ocean models with different fit parameters which makes it difficult to compare. Given that we compare SSA retrievals only for AOD_550nm > 0.2, we believe the difference in ocean model has minor impact for this comparison.

*Is the "BPDF scaling parameter" the same as the "Maignan scaling" parameter in Table 2? Recommend using one term consistently, and adding the Greek/Mathematical symbols corresponding to the written parameters in Table 2.*

We changed 'Maignan scaling parameter' to 'BPDF scaling parameter' in Table 2 and use that throughout the manuscript. Also, we added the symbol $B_{pol}$ to Table 2.

*Figures 19-21: the colorbar appears to be missing. What does the white line represent? Should "delta" be "\Delta" in Latex?*

We added the color bar in the revised version and clarify the meaning of the white circles: "The x-axis and y-axis have been divided in 100 bins. The white circles indicate the median y-value per x-bin. The color indicates the number of data points per bin."

Minor corrections:

*Line 142: lambda -> \lambda*

Corrected

*Line 201: om -> on*

Corrected

*Line 246 (and a few other places): correctly use ` as the opening single quote*

Corrected